# The Process-Mode-Driving Force of Cropland Expansion in Arid Regions of China Based on the Land Use Remote Sensing Monitoring Data

**Tianyi Cai [1,2], Xinhuan Zhang [1,\*], Fuqiang Xia [1], Zhiping Zhang [1,2], Jingjing Yin [1,2] and Shengqin Wu [1,2]**

1   State Key Laboratory of Desert and Oasis Ecology, Xinjiang Institute of Ecology and Geography, Chinese Academy of Sciences, Urumqi 830011, China; caitianyi14@mails.ucas.ac.cn (T.C.); xiafq@ms.xjb.ac.cn (F.X.); zhangzhiping15@mails.ucas.ac.cn (Z.Z.); yinjingjing11@mails.ucas.ac.cn (J.Y.); wushengqin16@mails.ucas.ac.cn (S.W.)
2   University of Chinese Academy of Sciences, Beijing 100049, China
\*   Correspondence: zhangxh@ms.xjb.ac.cn; Tel.: +86-099-1782-7314

**Abstract:** The center of gravity of China's new cropland has shifted from Northeast China to the Xinjiang oasis areas where the ecological environment is relatively fragile. However, we currently face a lack of a comprehensive review of the cropland expansion in oasis areas of Xinjiang, which is importantly associated with the sustainable use of cropland, social stability and oasis ecological security. In this study, the land use remote sensing monitoring data in 1990, 2000, 2010 and 2018 were used to comprehensively analyze the process characteristics, different modes and driving mechanisms of the cropland expansion in Xinjiang, as well as its spatial heterogeneity at the oasis area level. The results revealed that cropland in Xinjiang continued to expand from 5803 thousand hectares in 1990 to 8939 thousand hectares in 2018 and experienced three stages of expansion: steady expansion, rapid expansion, and slow expansion. The center of gravity of cropland showed the characteristic of shifting to the South. Edge expansion and encroachment on grassland were the dominant spatial pattern mode and land use conversion mode of Xinjiang's cropland expansion, respectively. The expansion of cropland in Xinjiang was affected by multiple factors. Irrigation conditions played a dominant role. Topography indirectly affected cropland expansion by affecting the suitability of agricultural production and development. Population growth and farmers' income were important driving forces. There was significant spatial heterogeneity in the intensity, mode and driving force of cropland expansion among different oasis areas in Xinjiang. The spatial shift of China's new cropland has occupied a large amount of water resources and ecological land in Xinjiang and exacerbated the vulnerability of the ecosystem in arid regions. The key to sustainable management of cropland in Xinjiang in the future lies in maintaining an appropriate scale of cropland and promoting the coordinated development of cropland, population, water resources and industry.

**Keywords:** cropland expansion; Xinjiang; land use conversion; landscape expansion index; Geodetector; Oasis



## 1. Introduction

As indicated by the latest World Population Prospects 2019 released by the United Nations, the global population is expected to reach 9.7 billion by 2050, and 11 billion by the end of this century [1]. A constantly increasing population and the ensuing growth in food consumption will pose unprecedented challenges to agriculture and cropland resources [2–4]. In this context, expanding the area of cropland has become one of the primary strategies for the supply side to satisfy the growing demand for food consumption on a global scale [5,6]. In the past three hundred years, the global cropland area has expanded by about five times, and most of the new cropland has come at the expense of forests and grasslands [7]. In the last seventy years, the frontier of global agricultural

expansion has shifted from Europe and North America to tropical regions [7–9], especially to South America [10–12] and sub-Saharan Africa [13–15].

As the most populous country in the world, China feeds nearly 18.7% of the world's population, despite possessing only 7.5% of its arable land [16], and thus China plays a significant role in global food security. With the rapid urbanization and industrialization process in China since the 21st century, more scholars have focused on the encroachment on cultivated land by construction land in the southeast coastal areas of China and its potential threats to China's food security [17–20], as these areas, featuring high levels of urbanization and industrialization and rapid development, are troubled by a prominent contradiction between cultivated land and construction land. The fact is, however, that the total area of cropland in China has remained relatively steady since 1990 and has not experienced a marked decrease. The reason for this is that while cropland has shrunk in southern China, it has continued to expand in northern China [21]. Furthermore, the focus of cropland reclamation in northern China has shifted from Northeast China and eastern Inner Mongolia to the oasis agricultural area in Xinjiang [21,22], and the expansion of cropland in the Xinjiang oasis area is still accelerating [23]. Xinjiang is located in the inland northwest of China and constitutes a major part of China's arid area, with an extremely fragile ecological environment [24]. Oasis is the landscape type with the highest ecologically sensitivity in arid regions, and it serves as the material basis for human survival and development in arid regions [25]. As the mainstay of modern oases, cropland is not only the material condition for safeguarding food security and promoting rural economic development, but also functions as an important protective screen in maintaining social stability and supporting ecological security [26,27]. In the face of the current triple stress of food security, ecological security and social security, it is urgent to gain a clear understanding of the spatial-temporal characteristics, different modes and formation mechanisms of cropland expansion in Xinjiang oasis areas, which will be of great significance for a renewed understanding of the profound impact of China's cropland pattern in a dynamic balance, and will also serve as an important basis for Xinjiang to implement the strategy of ecological civilization and promote a rational development and use of cropland.

With the continuous progress made in earth observation technology, remote sensing, due to its advantages of wide coverage in real time and at low cost, has emerged as the primary technical means for survey and monitoring of global land resources [21,28]. Based on remote sensing technology and its relevant products, fruitful research has been conducted on the change of cropland in Xinjiang. However, many scholars have studied the change of land use in Xinjiang [29–32], as well as its watershed [33–36] and the oasis unit scale [37–40], in which cropland was examined as only one of the many types of land use. There have also been scholars who directly studied the expansion of cropland at different geographic scales in Xinjiang. For example, Chen et al. analyzed the spatial-temporal changes and regional differences of cropland in Xinjiang in 1990, 2000, 2005, and 2008 based on interpretation of remote sensing data on land use [41]; Zhang et al. used comprehensive data of aerial photos, Landsat TM (thematic mapper) and Landsat OLI (operational land imager) and revealed the evolution of the modern oasis in the Sangong River Basin on the northern slope of the Tianshan Mountains since 1950 [42]; Zhang et al. explored the change of county-level cropland pattern in Xinjiang from 1988 to 2008 by use of land area data in the statistical yearbooks [43]. On the whole, all of the above studies concluded that the expansion of cropland in Xinjiang shows different characteristics in different phases in time and has continuously expanded in space from the alluvial plain with good water and soil resource conditions in the oasis to the desert beyond the oasis. However, the research period of these studies was basically limited to 2015 and earlier. As time passes, and the data are continuously updated, there is a rising need to understand the new characteristics of recent cropland changes. Therefore, it is necessary to provide updated data to fill the gap.

Regarding the mode of cropland expansion, most scholars have chosen to discuss this from the perspective of land use conversion. According to the research conclusions

available at present, scholars unanimously believe that encroachment on grassland and reclamation of unused land have been the main modes of cropland expansion in Xinjiang in the past [30,31,37]. Chen et al. also discussed the mode of cropland reclamation in Xinjiang from the perspective of soil texture and irrigation methods and concluded that the expansion of cropland has gradually shifted from loamy oasis areas to gravelly and sandy desert, from the diversion channel irrigation area to the groundwater irrigation area [44]. However, few scholars have studied the expansion mode of cropland in Xinjiang from the perspective of spatial patterns. In fact, it is important to have an in-depth understanding of the characteristics of extensive and intensive cropland expansion based on the association between spatial locations of the original cropland and the new cropland, which will be of great significance for an objective understanding of cropland expansion.

Regarding the driving forces of cropland expansion, scholars hold different opinions. Hu et al. believe that population growth has been the most important driving factor for the expansion of arable land in Xinjiang, while the government's agricultural subsidies and land reclamation policies have played supporting roles [45]. Zuo et al. also reached a similar conclusion in their research [46]. In comparison, the research of Chen et al. showed that the main factors promoting the expansion of cropland in Xinjiang were population growth and the restrictions from water and soil resources, but with the advancement of agricultural technology with respect to making use of water and soil, these restrictions are increasingly being weakened [41]. Zhu and Li believe that cropland expansion can be mainly attributed to the entities engaged in the agricultural business, who have tried to expand the scale of cash crops in order to pursue economies of scale [47]. Liu et al. found that temperature and precipitation had played a decisive role in the change of cropland by affecting the runoff in arid regions [31]. On the whole, these scholars focused on the socio-economic factors in their research of cropland expansion in Xinjiang. There is no denying that factors such as population, economy, policy, and technology do exert important influences. However, for arid areas where the ecological environment is relatively fragile, the role of natural environmental factors should not be underestimated. To fully understand the driving mechanism of Xinjiang's cropland expansion, it will be necessary to conduct a comprehensive assessment of its natural and socio-economic factors.

In consideration of the deficiencies of the above research, this study takes Xinjiang, China and its fifteen oasis areas as the study area and uses the four phases of land use remote sensing monitoring data in 1990, 2000, 2010 and 2018 to perform a systematic analysis of the phenomenon of cropland expansion. This paper aims to provide a reference for people to rationally understand the phenomenon of cropland expansion in Xinjiang and promote the reasonable development and use of its cropland. The main research objectives of this paper are to: (i) reveal the temporal and spatial characteristics of the expansion process of cropland in Xinjiang and its oasis areas; (ii) describe the mode of cropland expansion in Xinjiang and its oasis areas from the perspectives of spatial patterns and land-use conversion; (iii) comprehensively analyze how the factors of natural environment and socio-economy have served as the driving forces of cropland expansion in Xinjiang as well as its spatial heterogeneity; (iv) explore the ecological risks that may be brought by the cropland expansion in Xinjiang and propose suggestions for sustainable management of cropland in the future.

## 2. Materials and Methods

### 2.1. Study Area

Xinjiang is located in the central part of Eurasia and in the northwest of the People's Republic of China. It is located at 73°40′~96°23′ East longitude and 34°25′~49°10′ North latitude. With a typical temperate continental arid climate, it is the largest provincial administrative region in China, and also constitutes the principal part of the arid region in northwest China [24]. It has three major mountain systems including the Altai Mountains, the Tianshan Mountains, and the Kunlun Mountains, lying from North to South, two major

inland basins of Junggar and Tarim, forming a topographic pattern of "three mountains and two basins" and "mountain-oasis-desert" ecosystem landscape in general [24].

As of 2018, Xinjiang had 8.94 million hectares of arable land, which accounted for about 5% of the total land area of Xinjiang, and a total population of 24.87 million people, with an urbanization rate of 50.91% [48]. The number of employees in primary industry accounted for as much as 41% of the total number of employees in its three industries [48]. The oasis cropland in Xinjiang is not only the main spatial carrier of agricultural development, but also the production system most intimately related to oasis ecological security and rural social stability. Water is the lifeline of oasis agriculture and melt water from glaciers and mountain snow is the main water source for agricultural production in Xinjiang [49].

Oasis division is the division of a large oasis area into related oasis sub-systems through comprehensive integration, and one of the advantages of this division is that it comprehensively takes into consideration many aspects, including topographic patterns, water systems (watersheds or irrigation areas) and intactness of administrative division in the subregions [27]. In this study, the division scheme of Xinjiang Oasis [27] proposed by Hang was adopted to analyze the spatial heterogeneity of the cropland expansion in Xinjiang. The scheme divides Xinjiang into the Northern Xinjiang oasis area and the Southern Xinjiang oasis area. The Northern oasis area includes 6 oasis communities, namely Tacheng-Emin Valley Oasis area (TEVO), Tianshan North Slope West Oasis area (TNWO), Tianshan North Slope East Oasis area (TNEO), Irtysh-Ulungu River Oasis area (IURO), Ili Oasis area (ILO) and Aibi Lake Oasis area (ABLO). The Southern oasis area includes 9 oasis communities, namely Turpan Oasis area (TPO), Hami Oasis area (HMO), Aksu Oasis area (AKSO), Weigan River Oasis area (WGRO), Kashgar Oasis area (KSGO), Yerqiang Oasis area (YEQO), Kaidu-Kongque River Oasis area (KKRO), Hotan Oasis area (HTO) and Altun Mountain North Oasis area (AMNO). The specific spatial scope of each oasis area is shown in Figure 1.

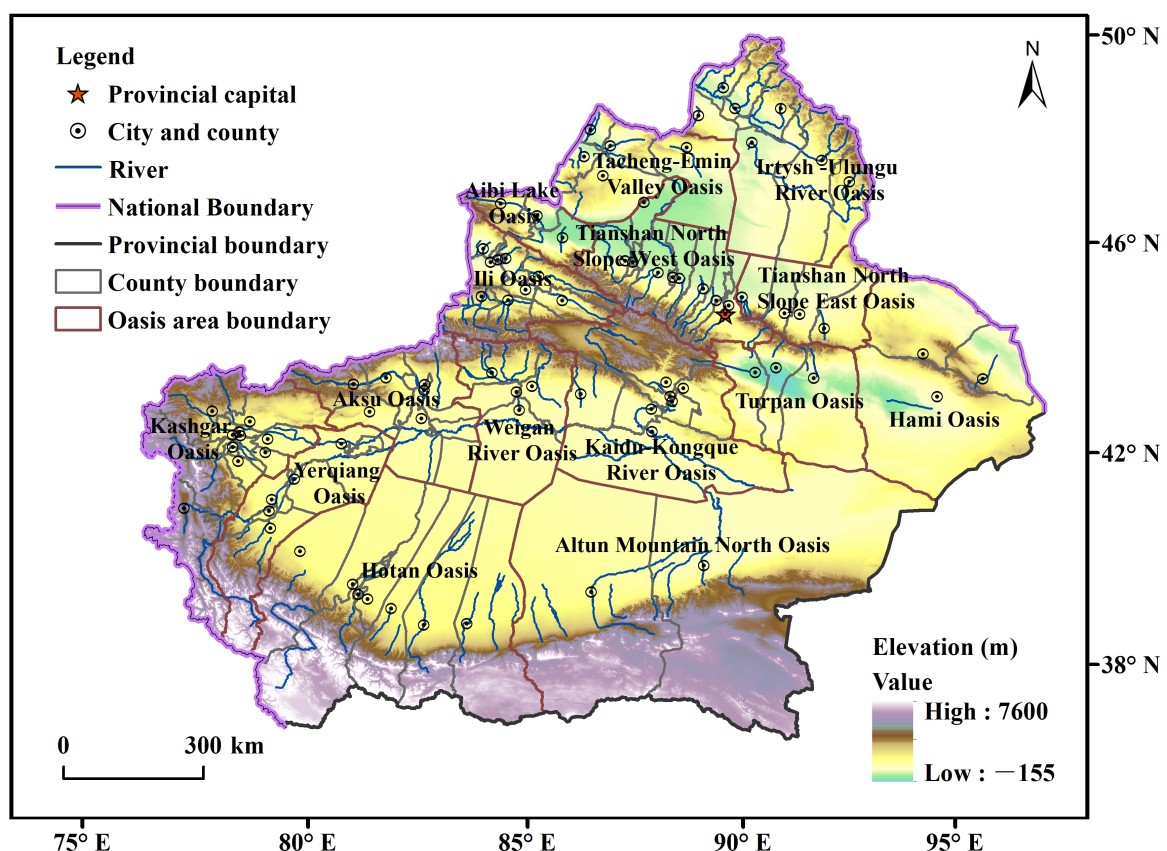

**Figure 1.** The geographic location of Xinjiang and its oasis divisions.

### 2.2. Methods

In this study, a research framework was constructed to analyze the phenomenon of cropland expansion in Xinjiang based on "process-mode-driving force". The specific research framework and technical route are shown in Figure 2.

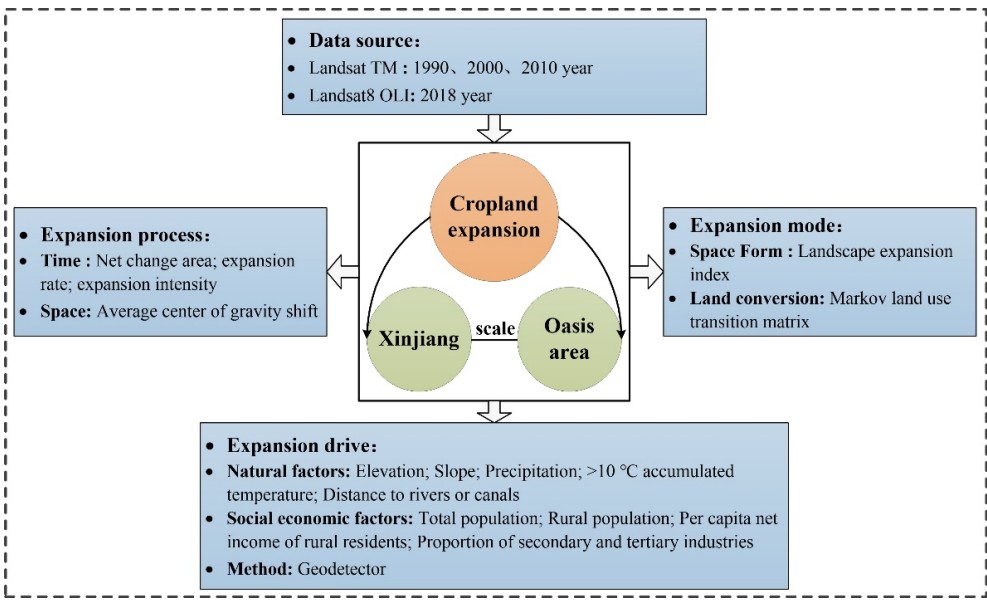

**Figure 2.** Research framework and technical route.

2.2.1. Method for Describing the Process of Cropland Expansion

Learning from the methods of Liu and Ning [21,23], this study used cropland net change area (CLNCA), cropland expansion rate (CLER) and cropland expansion intensity (CLEI) to describe the characteristics of the temporal process of cropland expansion, and used the center of gravity of the spatial distribution of cropland (CGSDC) and the trajectory of its shift to describe the characteristics of the spatial process of cropland expansion.

CLNCA ($A_{i,t}$) is an indicator reflecting the changed scale of cropland in a certain period of time, and the calculation formula is:

$$A_{i,t} = A_{i,t2} - A_{i,t1} \tag{1}$$

In the formula, $A_{i,t1}$ and $A_{i,t2}$, respectively, represent the area of cropland in the study area at the beginning and the end of the study period from $i$ to $t$. When the value of $A_{i,t}$ is positive, the larger the value, the larger the scale of cropland expansion during the period; when it is negative, the opposite is true.

CLER ($V_{i,t}$) is an indicator reflecting the speed of cropland expansion in the same research unit in different periods. The calculation formula is:

$$V_{i,t} = (A_{i,t2} - A_{i,t1}) / A_{i,t1} \times (1/t) \times 100\% \tag{2}$$

In the formula, when $V_{i,t}$ is greater than 0, it indicates that the cropland in the area is increasing, and when it is less than 0, the cropland in the area is decreasing. When $V_{i,t}$ is greater than 0, the larger the value, the faster the expansion of cropland.

CLEI ($D_{i,t}$) is the ratio of the cropland expansion scale of a certain research unit to the overall cropland expansion scale of the entire research area. This index excludes the influence from different scales of cropland in different research units, and can be used to analyze different intensities of cropland expansion of different research units in the same period. The calculation formula is:

$$D_{i,t} = |A_{i,t2} - A_{i,t1}| / |A_{t2} - A_{t1}| \times 100\% \tag{3}$$

In the formula, $A_{t1}$ and $A_{t2}$ represent the total cropland area of the study area at the beginning and the end of the period, respectively. The larger the value of $D_{i,t}$, the more intense the change of cropland area in the subregion.

CGSDC refers to the point through which the force of gravity will pass; however, cropland is placed in the gravitational field. In this study, the weighted mean center of gravity is used to describe the spatial macro movement process of cropland in different periods [46]. The calculation formula of mean center of gravity is:

$$M(\overline{X}, \overline{Y}) = \left| \frac{\sum_{i=1}^{n} w_i x_i}{\sum_{i=1}^{n} w_i} \times \frac{\sum_{i=1}^{n} w_i y_i}{\sum_{i=1}^{n} w_i} \right| \tag{4}$$

In the formula, $x_i$ and $y_i$ are the latitude and longitude coordinates of the $i$th subregion; $w_i$ is the weight of the subregion; $n$ is the number of subregions. To improve the calculation accuracy of the mean center of gravity, this study made use of ArcGIS software (West Redlands, CA, USA) and used Xinjiang as a mask to generate a grid with squares of 10 km × 10 km. On this basis, the area of cropland in each grid square was calculated. Then the grid square was deemed to be a subregion, and the area of cropland in the grid square as the weight, and thus the weighted mean center of gravity could be calculated.

### 2.2.2. Method for Describing the Mode of Cropland Expansion

This study analyzed the mode of cropland expansion mainly from two perspectives: spatial patterns and land use conversion. Spatial patterns in cropland expansion were described by the landscape expansion index (LEI) of the cropland, while land use conversion was described by the Markov land use transition matrix.

LEI is an approach for identifying the mode of urban spatial expansion by determining the relationship between the spatial locations of existing urban land and that of the newly added urban land [50]. This method is now mainly used to describe the expansion mode of urban space. Due to its strong dependence on water resources, oasis cropland has a spatially concentrated distribution characteristic [27], which provides the possibility of carrying out research on the expansion of oasis cropland in terms of spatial form. This study followed the method of Ouyang and Zhu of identifying spatial patterns in cropland expansion [51]. We divided the spatial patterns of cropland expansion into three types: infilling, edge expansion and outlying. The infilling expansion means the new cropland is formed by filling the gaps within the existing cropland space; the edge expansion refers to the extension and expansion of the new cropland along the edge of the existing cropland; the outlying expansion means that the new cropland is isolated from the existing cropland and that they are not adjacent to each other in terms of their spatial locations. The calculation formula of the LEI is:

$$LEI = L_{com}/P_{new} \tag{5}$$

In the formula, $L_{com}$ is the length of the common boundary shared by the new cropland and the existing cultivated land; $P_{new}$ is the perimeter of the new cropland. When $LEI = 0$, it is the outlying type; when $0 < LEI \leq 0.5$, it is the edge expansion type; when $0.5 < LEI \leq 1$, it is the infilling type.

The Markov land use transition matrix is derived from the quantitative description of the state of the system and the transition of its state in a system analysis and has been widely used in current land use change and simulation analyses [21,31,33,37]. This method can accurately reveal the conversion process between cropland and other land use types, as well as the area and spatial location of the conversion. Therefore, this method can be used to characterize the mode of cropland expansion from the perspective of land type conversion. The specific calculation formula is:

$$P_{gain(i),j} = (P_{i,j} - P_{j,i})/(P_{i.} - P_{.i}) \times 100, \, i \neq j \tag{6}$$

In the formula, $P_{gain(i),j}$ is the proportion of land use type *i* converted to land use type *j* in the net increased area of all the land use types in the *i*th row of the transition matrix, i.e., the contribution rate of the conversion; $P_{i,j}$ and $P_{j,i}$ are single values in the transition matrix table; $P_{i.}$ is the area of a land use type of the *i*th row at the end of the period, and $P_{.i}$ is the area of the land use type at the beginning of the period.

### 2.2.3. Methods to Analyze the Driving Forces of Cropland Expansion

- Geodetector

Geodetector is a statistical method for detecting spatially stratified heterogeneity and revealing the driving factors behind it [52,53]. Its core idea is: if an independent variable has an important influence on a dependent variable, then the spatial distribution of the independent variable and the dependent variable should be similar. This method does not require linear assumptions, and has an elegant form and clear physical meaning. In addition, it can well reveal the explanatory power of natural and social economic factors that lead to the spatial differentiation of dependent variables [52]. This method was initially applied to the study of risks of endemic disease and relevant geographic influencing factors [53]; later, it began to be widely used to study the formation mechanisms of urbanization [54], the driving forces of land use change [55], influencing factors of carbon emission [56], and so on. Specifically, it can be expressed as:

$$q = 1 - \frac{\sum_{h=1}^{L} N_h \sigma_h^2}{N \sigma^2} \tag{7}$$

In the formula, *q* is a measure of the explanatory power of the independent variable; *L* is the stratification of the dependent variable or independent variable; $N_h$ and $\sigma_h^2$ are the number of units and variance of the *h*th stratum, respectively; *N* and $\sigma^2$ are the number of units and variance of the entire study area, respectively. Assuming that $\sigma_h^2 \neq 0$, the model is valid, and the value of *q* is in the interval [0, 1]. When *q* = 0, it indicates that the dependent variable is randomly distributed. The larger the value of *q*, the greater the influence of the stratification factor on the dependent variable. Please refer to the website http://www.geodetector.cn/ (accessed on 20 May 2021) for the specific theory and implementation process of this model.

- Selection of influencing factors and definition of variables

With respect to the driving forces of cropland expansion, previous studies have mainly focused on the impact of social and economic factors such as population growth, economic development, technological progress, policy impetus, and profit impetus [31,41,45–47]. Some scholars also stressed the influences of climate change [57], topography [22], surface runoff [31], and other natural factors. On the basis of past research, and also taking into consideration the characteristics of the oasis areas in Xinjiang in terms of natural background and socio-economic development, as well as the availability of data, this study started from dimensions of both the natural environment and social economy to select six factors, which included topography, hydrothermal conditions, irrigation conditions, population growth, farmers' income increase, and industrial structure, to analyze their influences on the expansion of cropland (Figure 3). Specifically, the factor of topography, described by the two indicators elevation (ELV) and slope (SP), indirectly affect cropland expansion by influencing the combined features of surface water and heat as well as the difficulty of cropland development. Water and heat conditions, described by the two indicators annual average precipitation (AAP) and >10 °C accumulated temperature (AT10), have an important influence on agricultural production activities and their spatial distribution, and then indirectly affect the development and use of cropland. Irrigation conditions are the basic guarantee for the development of oasis agriculture. Agricultural irrigation is realized by the diversion of water from natural rivers or artificial canals, and is described by the spatial distance to rivers or canals (DRC). Population growth indirectly promotes the expansion of cropland. For one thing, it brings increasing demand for food;

for another, a growing agricultural population means increasing demand for employment. Population growth is described by the two indicators total population (TPOP) and rural population (RPOP). Farmers want to expand the area of arable land so that they can produce more agricultural products to sell in the market and gain more economic income. This is the economic motivation that drives them to expand cropland. The increase in farms' income is described by per capita net income of rural residents (PNIRR). Industrial structure exerts an important impact on the land use structure of a region. When the primary industry takes up a high proportion, the intensity of development and use of agricultural land will also be high. Industrial structure is described by proportion of secondary and tertiary industries (PSTI).

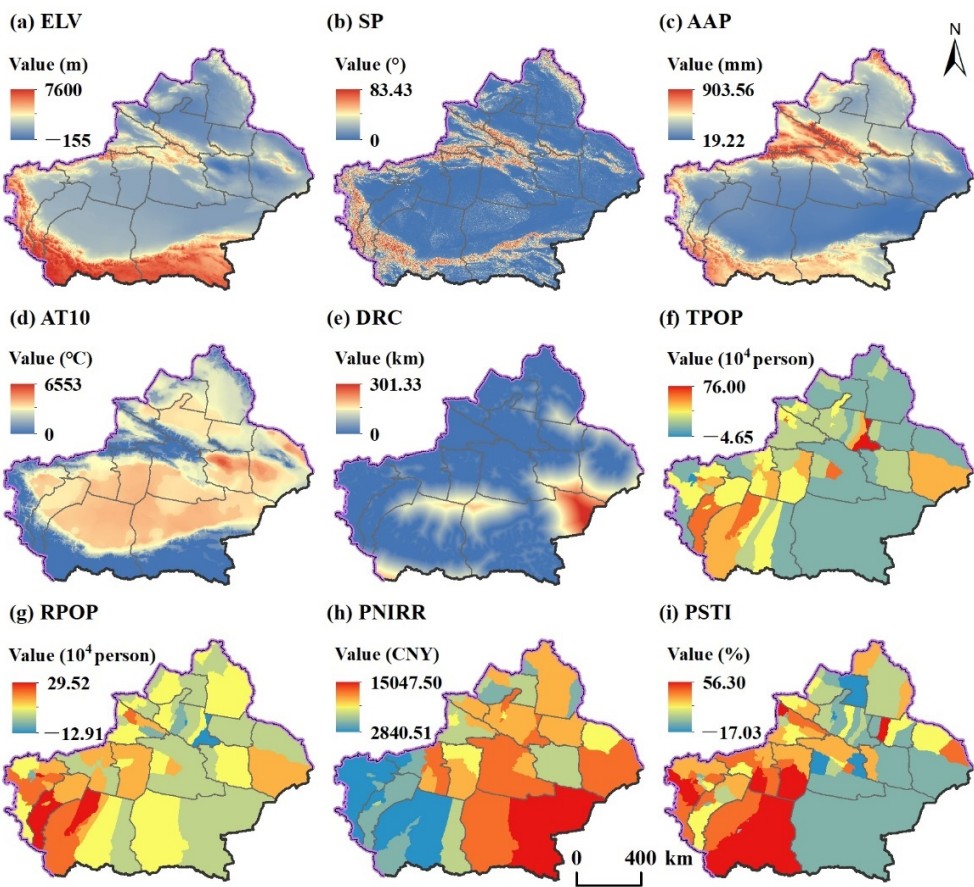

**Figure 3.** Spatial distribution of main factors affecting cropland expansion in Xinjiang (**a–i**). (Note: The values of the four indicators including TPOP, RPOP, PNIRR, and PSTI are their respective amount of changes from 1990 to 2018).

## 2.3. Data Sources and Processing

Table 1 shows the data source and description of this study. Data on cropland come from the land use remote sensing monitoring data set released by the Resource and Environment Science and Data Center of the Chinese Academy of Sciences (RESDC) (http://www.resdc.cn, accessed on 20 May 2021), with a spatial resolution of 30 m and including land use data of 1990, 2000 and 2010 based on Landsat TM [21], and the newly established land use data set of 2018 based on Landsat8 OLI (Figure 4) [23]. The DEM data come from Geospatial Data Cloud (http://www.gscloud.cn, accessed on 20 May 2021), with a spatial resolution of 30 m, and the slope data are extracted from the DEM data with the help of ArcGIS 10.7 software (West Redlands, CA, USA). The data of AT10 and AAP are from the China meteorological background data set released by the RESDC, with a spatial resolution of 500 m, and were resampled to a resolution of 30 m for our use. The vector data of administrative boundaries and rivers and canals come from the 1:1 million Chinese

basic geographic database provided by the China National Catalogue Service for Geographic Information (http://www.webmap.cn/, accessed on 20 May 2021). The distance to rivers and canals was obtained using the distance analysis tool of ArcGIS 10.7 software (West Redlands, CA, USA). Socio-economic statistics come from the Xinjiang Statistical Yearbook and the statistical yearbooks of various prefectures in Xinjiang. The net income of rural residents was converted into sums comparable with 1990 as the base period in accordance with the consumption price index of rural residents in the corresponding year.

**Table 1.** Datasets used in this research. All website accessed on 20 May 2021.

| Data | Data Sources | Year (s) | Resolution |
|---|---|---|---|
| Land use data | http://www.resdc.cn | 1990, 2000, 2010, 2018 | 30 m |
| DEM data | http://www.gscloud.cn | — | 30 m |
| AAP and AT10 data | http://www.resdc.cn | 1990–2015 | 500 m |
| Administrative boundaries data | http://www.webmap.cn/ | 2015 | — |
| Rivers and canals data | http://www.webmap.cn/ | 2015 | — |
| Socio-economic statistics data | Statistical Yearbooks of Xinjiang and its various prefectures | 1990, 2000, 2010, 2018 | — |

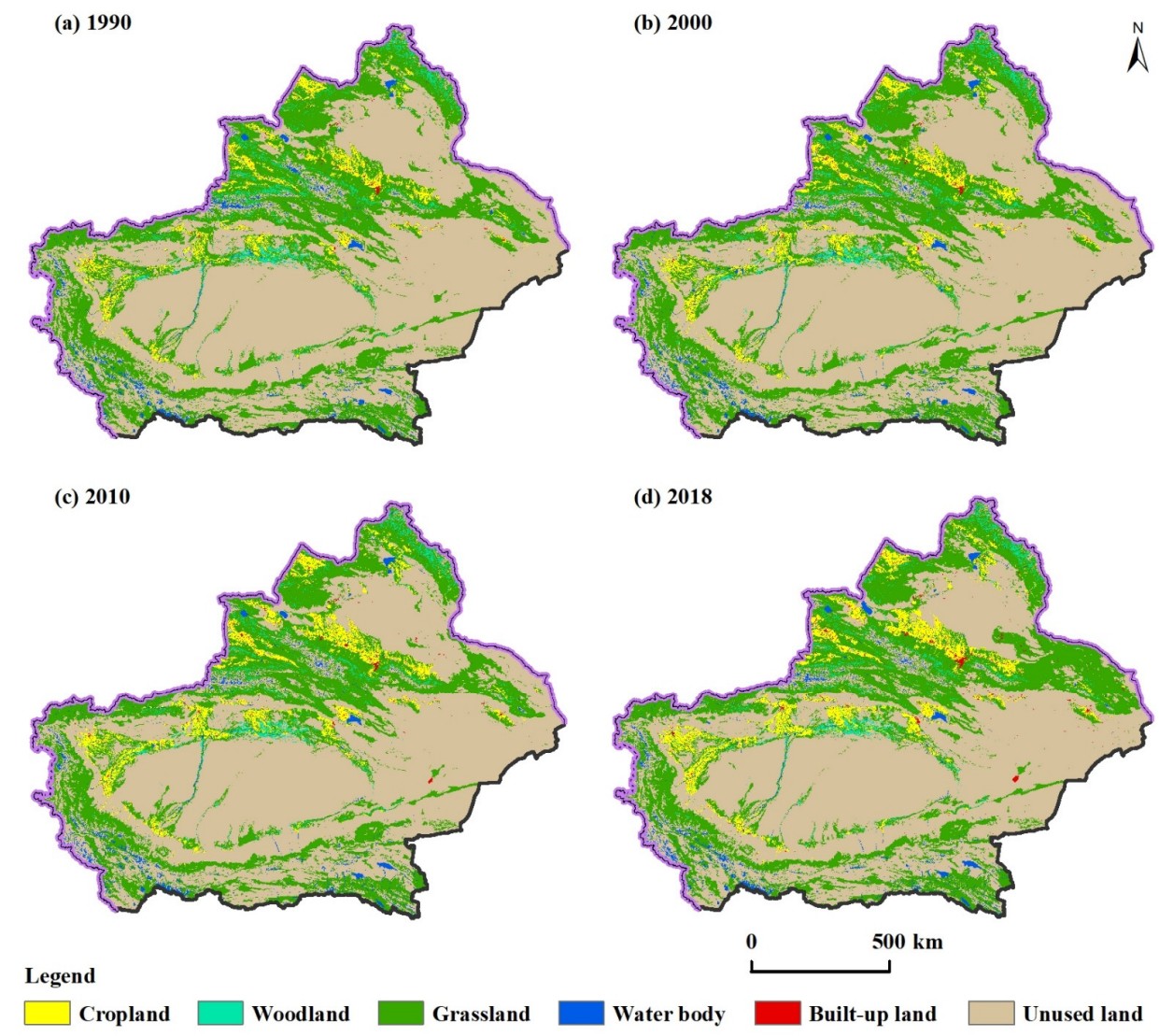

**Figure 4.** Land use types and spatial distribution in Xinjiang from 1990 to 2018 (**a–d**).

## 3. Results

*3.1. The Process of Cropland Expansion*

3.1.1. Temporal Process

Cropland in Xinjiang showed an obvious expansion in scale during the study period and underwent three stages of change: steady expansion, rapid expansion, and slow expansion (Table 2). From 1990 to 2000, the CLER was 1.32%, showing a trend of steady expansion; from 2000 to 2010, the CLER reached 2.28%, showing a trend of rapid expansion; from 2010 to 2018, the CLER was 1.36%, which was slower than that of the previous period. In general, the area of cropland in Xinjiang continued to increase from 5802.89 thousand hectares in 1990 to 8938.54 thousand hectares in 2018, which means a net increase of 3135.65 thousand hectares and CLER of 1.93%.

**Table 2.** Changes in the quantity of cropland in different periods of Xinjiang during 1990–2018.

| Year | Area of Cropland ($10^3$ hm$^2$) | Period | CLNCA ($10^3$ hm$^2$) | CLER (%/year) |
|---|---|---|---|---|
| 1990 | 5802.89 | 1990–2000 | 765.31 | 1.32 |
| 2000 | 6568.20 | 2000–2010 | 1495.27 | 2.28 |
| 2010 | 8063.47 | 2010–2018 | 875.07 | 1.36 |
| 2018 | 8938.54 | 1990–2018 | 3135.65 | 1.93 |

Judging from the performance of different oasis areas in Xinjiang (Figure A1 in Appendix A and Table 3), there was obvious spatial heterogeneity in the change of cropland scale. In terms of the rate of change in the scale of cropland, except for the period 2010–2018, when TPO experienced a reduction in the area of cropland, during the rest of the periods, the area of cropland in all oasis areas showed a general trend of expansion. Among them, three oasis areas (KSGO, HTO, and AMNO) experienced an accelerated rate of cropland expansion consistently, while the remaining twelve oasis areas largely shared the same phasal characteristics with Xinjiang in terms of CLER. On the whole (1990–2018), AMNO registered the fastest expansion of cropland (7.07%/year). In addition, IURO, ABLO, KKRO, AKSO, TNWO and HMO also had faster cropland expansion than the average rate of Xinjiang during the study period, and TPO had the slowest expansion (0.67%/year). Generally speaking, the CLER of the Southern Xinjiang oasis areas (2.10%/year) was faster than that of the Northern Xinjiang oasis areas (1.77%/year).

**Table 3.** Temporal and phasal characteristics of cropland expansion in different oasis areas in Xinjiang during 1990–2018.

| Name of Oasis Areas | 1990–2000 | | 2000–2010 | | 2010–2018 | | 1990–2018 | |
|---|---|---|---|---|---|---|---|---|
| | CLEI (%) | CLER (%/year) | CLEI (%) | CLER (%/year) | CLEI (%) | CLER (%/year) | CLEI (%) | CLER (%/year) |
| TEVO | 1.17 | 0.21 | 6.49 | 2.22 | 1.95 | 0.40 | 3.92 | 1.03 |
| TNWO | 14.82 | 1.11 | 29.07 | 3.83 | 11.47 | 0.80 | 20.68 | 2.27 |
| TNEO | 4.78 | 1.12 | 4.42 | 1.82 | 1.49 | 0.38 | 3.69 | 1.27 |
| IURO | 7.91 | 2.36 | 8.30 | 3.91 | 9.56 | 2.37 | 8.55 | 3.73 |
| TPO | 1.86 | 1.29 | 0.62 | 0.74 | 0.33 | −0.27 | 0.66 | 0.67 |
| HMO | 2.94 | 2.10 | 2.83 | 3.26 | 0.18 | 0.12 | 2.12 | 2.21 |
| ILO | 7.67 | 0.78 | 4.81 | 0.89 | 3.87 | 0.48 | 5.24 | 0.78 |
| ABLO | 4.18 | 1.84 | 5.44 | 3.95 | 3.53 | 1.34 | 4.60 | 2.96 |
| AKSO | 14.85 | 2.33 | 11.25 | 2.80 | 10.37 | 1.47 | 11.88 | 2.73 |
| WGRO | 6.53 | 1.17 | 3.61 | 1.13 | 14.01 | 2.89 | 7.23 | 1.90 |
| KSGO | 5.24 | 0.86 | 4.04 | 1.19 | 17.33 | 3.33 | 8.04 | 1.92 |
| YEQO | 10.78 | 1.53 | 6.71 | 1.61 | 6.37 | 0.96 | 7.61 | 1.57 |
| KKRO | 12.08 | 2.46 | 8.41 | 2.69 | 10.33 | 1.90 | 9.84 | 2.94 |
| HTO | 3.70 | 0.94 | 2.84 | 1.28 | 7.06 | 2.06 | 4.23 | 1.56 |
| AMNO | 1.49 | 4.23 | 1.16 | 4.53 | 2.8 | 5.51 | 1.70 | 7.07 |

In terms of the intensity of the change in the scale of cropland (Table 3), IURO showed a continuous increase in the CLEI, while TPO, HMO, TNEO, ILO, AKSO and YEQO showed a continuous decrease, and the remaining eight oasis areas had the most intense cropland expansion during 2000–2010. On the whole (1990–2018), TNWO had the most intense cropland expansion, with an expansion intensity of 20.68%. Other oasis areas, including AKSO, KKRO, IURO and KSGO, were also among the Top 5 in terms of CLEI. The cumulative CLEI of these five oasis areas reached as high as 59%, showing the characteristic of strong regional concentration. Generally speaking, the CLEI in the Southern Xinjiang oasis areas (53.31%) was higher than that in the Northern Xinjiang oasis areas (46.69%).

### 3.1.2. Spatial Process

Judging from the scale of Xinjiang (Figure 5), from 1990 to 2018, the shift of CGSDC in Xinjiang showed in its trajectory a characteristic of "slightly shifting toward southeast-leaping forward to northeast-moving back to southwest". Although the area of cropland continued to expand, its proportion in both southeast half and northwest half of Xinjiang (as divided by the "Qitai-Qira" line) stayed largely stable at 1:9, showing strong dynamic stability.

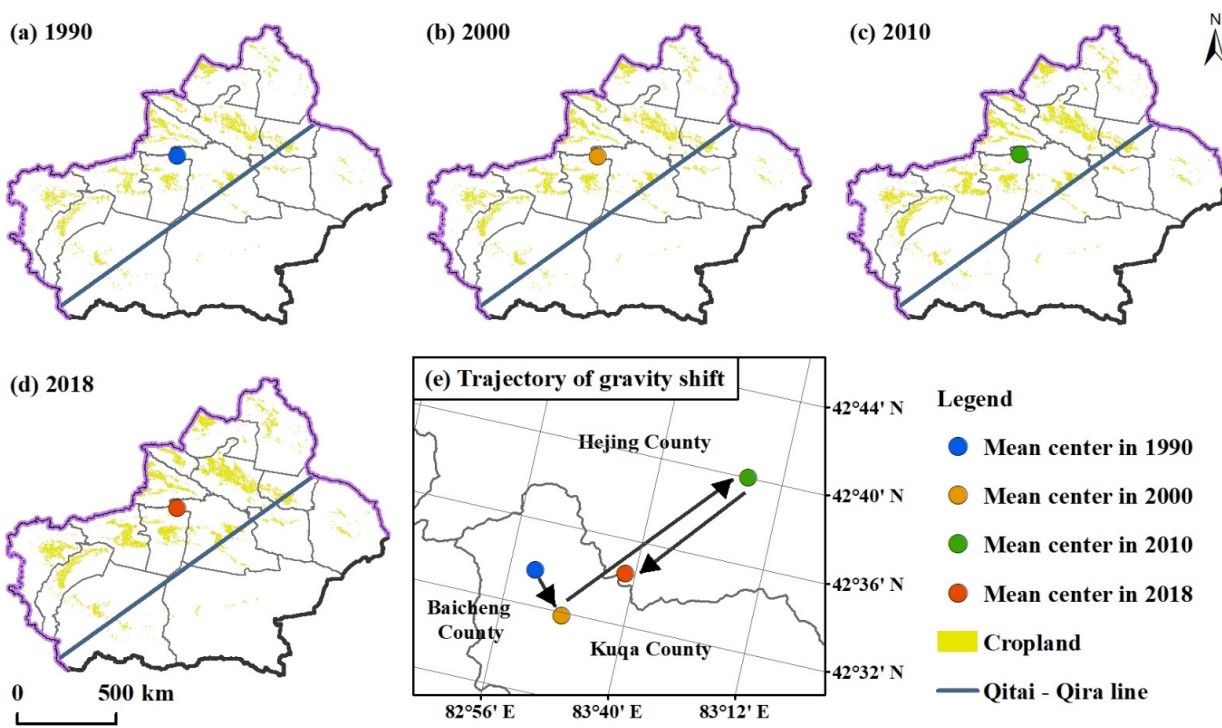

**Figure 5.** Spatial distribution of cropland (**a**–**d**) in Xinjiang and the trajectory of the shift of center of gravity (**e**) during 1990–2018.

Specifically, the CGSDC in Xinjiang was located at the northwestern end of Kuqa County, Aksu Prefecture in 1990 (82.96°E, 42.56°N). In 2000, the CGSDC moved slightly to the southeast by 4.85 km and moved quickly to the northeast by 20.45 km in 2010. By 2018, the CGSDC moved back toward the southwest (by 13.71 km) to the junction of Hejing County and Kuqa County (83.06°E, 42.57°N) again. According to the method of Deng [58], the line connecting Qitai County and Qira County in Xinjiang could be used to roughly divide Xinjiang into the northwest half and the southeast half. The land space of the two regions accounted for 54% and 46% of Xinjiang, respectively; however, they had a huge difference in terms of cropland scale. On the one hand, the area of cropland on both sides of the line continued to expand (from 1990 to 2018, the area of cropland in the northwestern half continued to increase from 5236.39 thousand hectares to 8042.99 thousand hectares,

and the area of cropland in the southeast half continued to increase from 566.50 thousand hectares to 895.55 thousand hectares). On the other hand, the two regions divided by the line kept their cropland scale largely in a dynamic balance, with a stable ratio of cropland on the two sides remaining at around 9:1 (the proportion of cropland in the northwest half was 90.24%, 89.55%, 89.69% and 89.98% in the four cross-sections of time, respectively).

Judging from the scale of oasis areas (Figure 6), significant differences existed among the oasis areas in their spatial expansion of cropland. From the perspective of the trajectory of shift of the CGSDC, generally three types could be identified. The first type was a trajectory approximating a C-shaped trend of change. Oasis areas of this type included WGRO, KSGO, AKSO, KKRO, TNEO, HMO, IURO and TPO. The second type is a trajectory approximating a linear trend of change. Oasis areas of this type included TNWO, TEVO and ABLO. The third type was a trajectory showing a circuitous trend of change, in which the cropland expanded in disorder. Oasis areas of this type included ILO, AMNO and YEQO. From the perspective of the shift distance of the CGSDC during different periods, four oasis areas (HMO, KKRO, HTO and AMNO) had the longest shift distance during 1990–2000, and six oasis areas (TEVO, TNWO, ILO, ABLO, AKSO and YEQO) had the longest shift distance during 2000–2010, while the remaining five oasis areas had the longest shift distances during 2010–2018.

### 3.2. Modes of Cropland Expansion
#### 3.2.1. Spatial Patterns

It can be seen from Table 4 and Figure 7 that the dominant spatial pattern of cropland expansion in Xinjiang during the study period was edge expansion, while other types of outlying and infilling also existed. Edge expansion was the dominant spatial pattern of cropland expansion in Xinjiang, and its contribution to the new cropland was 67.00%, 63.40% and 59.42% in the three periods, respectively, showing a significant path dependence of Xinjiang in its cropland expansion. The new cropland was mainly distributed around the edges of the original cropland, presenting the characteristic of "center-periphery" agglomeration in its distribution, but this type of expansion made a decreasing contribution to new cropland. Outlying expansion is another important spatial pattern of cropland expansion in Xinjiang, and its contribution to new cropland was 19.74%, 20.02% and 19.81% in the three periods, respectively. This shows that there was a certain level of jump diffusion in the cropland expansion in Xinjiang. Affected by various factors such as new irrigation water sources, resettlement and comprehensive agricultural development, the new cropland and the original cropland were isolated in space, forming new independent growth points. This type of expansion had a relatively stable proportion. Infilling expansion was also an important spatial pattern of cropland expansion in Xinjiang, and its contribution to new cropland was steadily on the increase (its proportions in the three periods was 13.26%, 16.59%, and 20.76%, respectively). This shows that the cropland expansion in Xinjiang had a certain degree of agglomeration. The internal gaps of the original cropland were constantly filled, which increased the concentration of cropland and steadily expanded its scale in space. It was mainly a synthetic result from the comprehensive improvement projects, reclamation of abandoned rural settlements, demand of agricultural mechanization operation and other factors.

**Table 4.** Statistics of spatial patterns of cropland expansion in Xinjiang during 1990–2018.

| Period | Types of Spatial Patterns of Cropland Expansion (%) | | |
|---|---|---|---|
| | Outlying Expansion | Edge Expansion | Infilling Expansion |
| 1990–2000 | 19.74 | 67.00 | 13.26 |
| 2000–2010 | 20.02 | 63.40 | 16.59 |
| 2010–2018 | 19.81 | 59.42 | 20.76 |

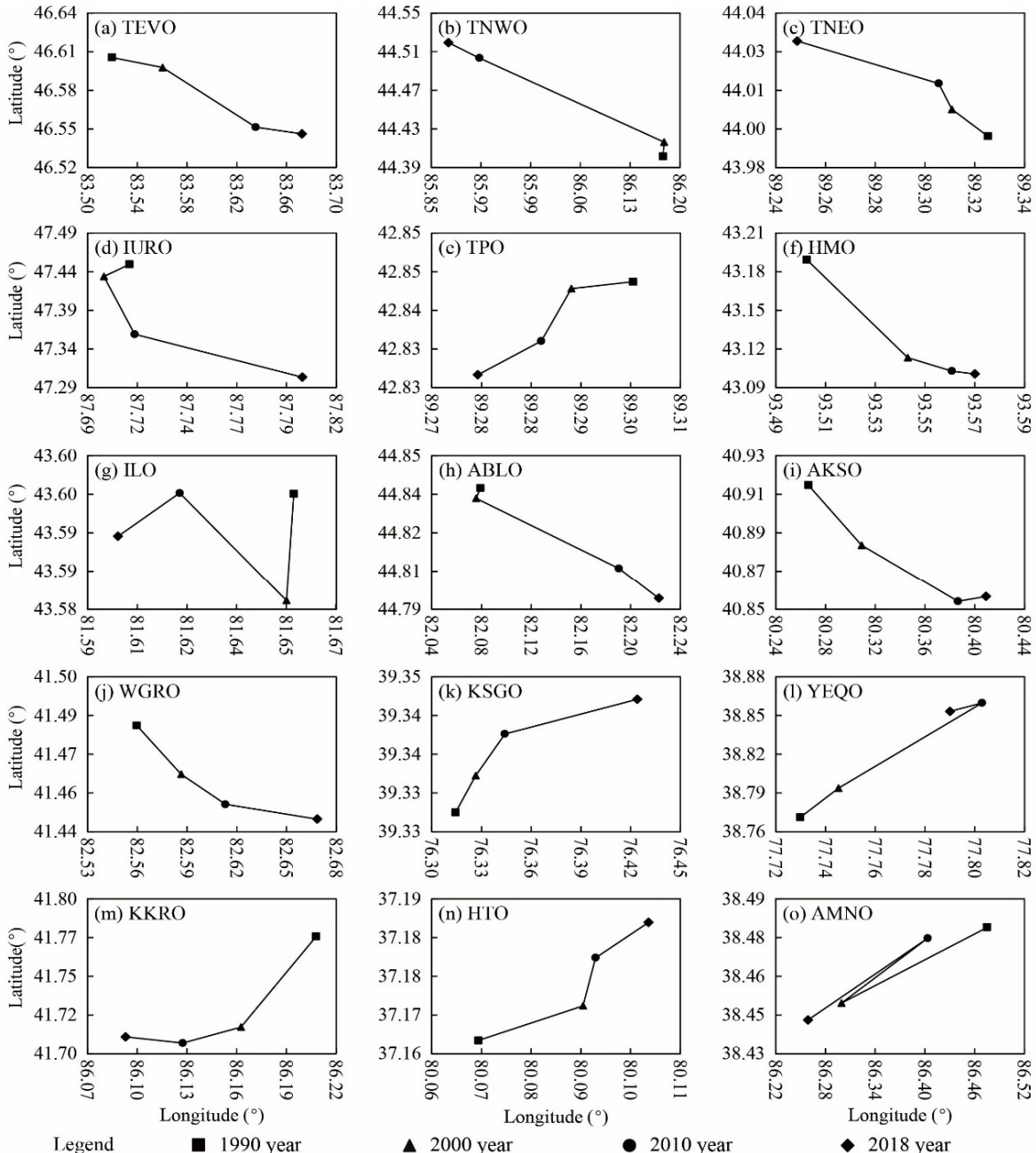

**Figure 6.** The trajectory of the shift of the center of gravity of cropland in various oasis areas in Xinjiang during 1990–2018 (**a**–**o**).

Regarding the contribution rates of different spatial patterns to the new cropland in different oasis areas, significant differences existed (Figure 7). Edge expansion made prominent contributions to cropland expansion in TNWO, AKSO and KKRO during the period 1990–2010, and its contribution rate in TNWO during 2000–2010 appeared to be polarized. During 2010–2018, in addition to the above three oasis areas, its contribution rate in KSGO and WGRO increased significantly. Outlying expansion made prominent contributions to cropland expansion in the TNWO, ILO and KKRO during the period 1990–2000. It made relatively big contributions to cropland expansion in TNWO and IURO during 2000–2010 and registered the highest contribution rate in IURO during 2010–2018, showing an obvious characteristic of polarization. Infilling expansion made significant contributions to cropland expansion in TNWO, YEQO and KSGO during 1990–2000. During 2000–2010, it made the biggest contribution in TNWO, and similar contributions in other oasis areas.

During 2010–2018, the gap among its contribution rates in different oasis areas was further widened, and generally the contribution rates of TNWO and KSGO are more prominent.

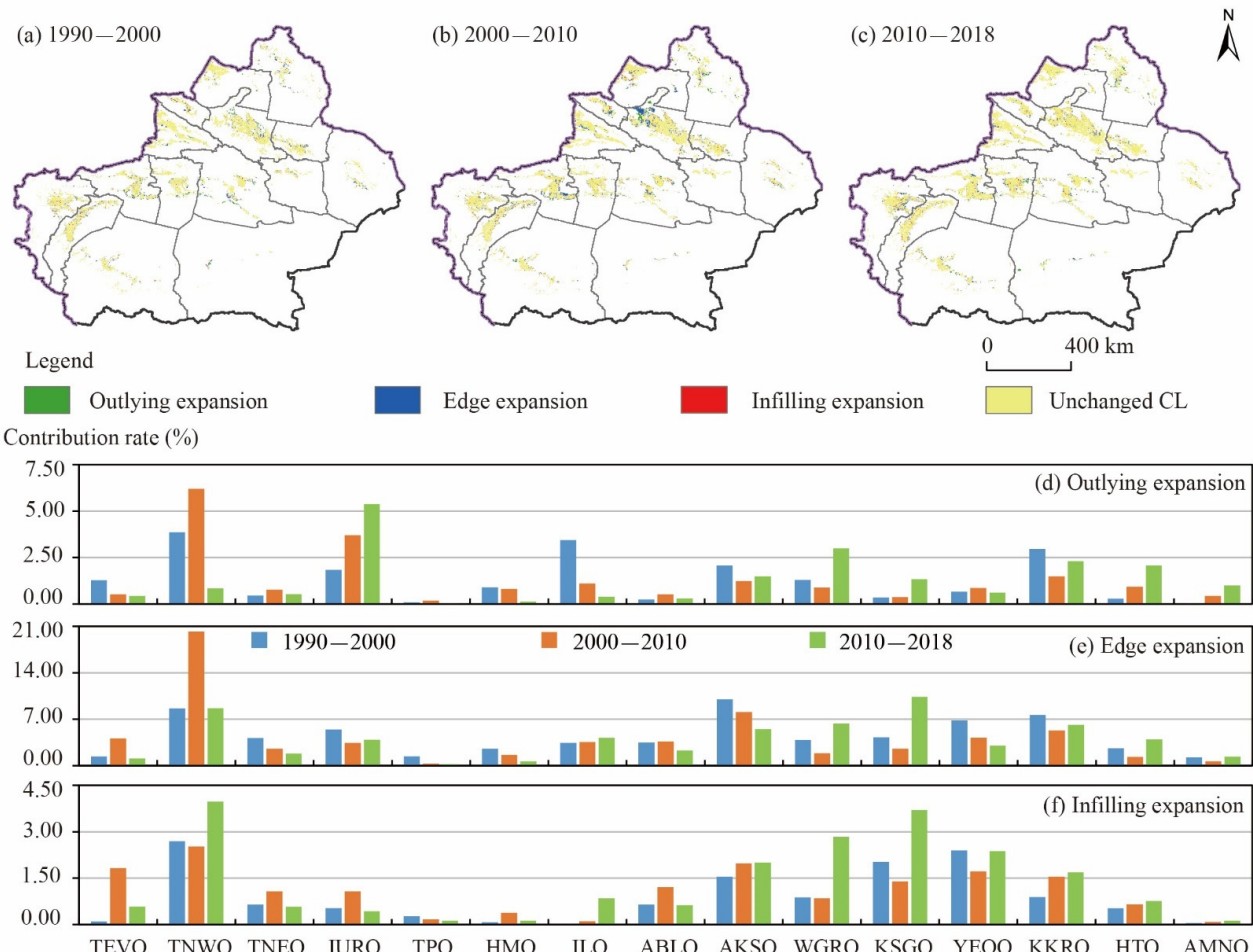

**Figure 7.** The spatial distribution of the spatial patterns of cropland expansion in Xinjiang (**a**–**c**) and the changes in the contribution rate of each oasis area to the newly added cropland (**d**–**f**) during 1990–2018.

Regarding the evolution of spatial patterns of cropland expansion in different oasis areas (Figure 8), obvious regional differences also existed. In this study, a spatial pattern was determined to be the dominant mode of expansion in a period when the area attributable to this spatial pattern accounted for more than 33.33% of the total area. In general, spatial patterns evolved in the following two scenarios. The first scenario was that edge expansion was the dominant mode throughout the three periods of the study, and the new cropland continuously spread and grew based on the original cropland. This scenario was the case for the largest number of oasis areas including TNWO, TNEO, HMO, ABLO, AKSO, KSGO, KKRO and WGRO. The second scenario was that the dominant spatial pattern of cropland expansion experienced changes over different study periods. TEVO, ILO, IURO, HTO, AMNO, TPO and YEQO were such examples.

### 3.2.2. Land Use Conversion Patterns

It can be seen from Table 5 and Figure 9 that the expansion of cropland in Xinjiang during the study period was evolving from a single mode of land use conversion dominated by grassland occupation to a dual development mode dominated by grassland occupation and reclamation of unused land. Grassland occupation was the major mode of land use conversion for the expansion of cropland in Xinjiang. The contribution rates of grassland to new cropland in the three periods were 92.04%, 88.91% and 76.58%, respec-

tively. This shows that cropland in Xinjiang was expanded at the expense of large amounts of ecological land. However, with strengthened ecological protection of grassland and the implementation of the policy of returning farmland to grassland, the proportion of new cropland through occupation of grassland was gradually decreasing. With the technological development made in water and soil development, as well as farmland water-saving irrigation, the reclamation of unused land grew to become another important mode of land use conversion for cropland expansion in Xinjiang, and its contribution rates were 6.75%, 10.20% and 21.02% in the three periods, respectively, showing a trend of continuous growth. Cropland converted from forest land, water area and construction land took up relatively small proportions.

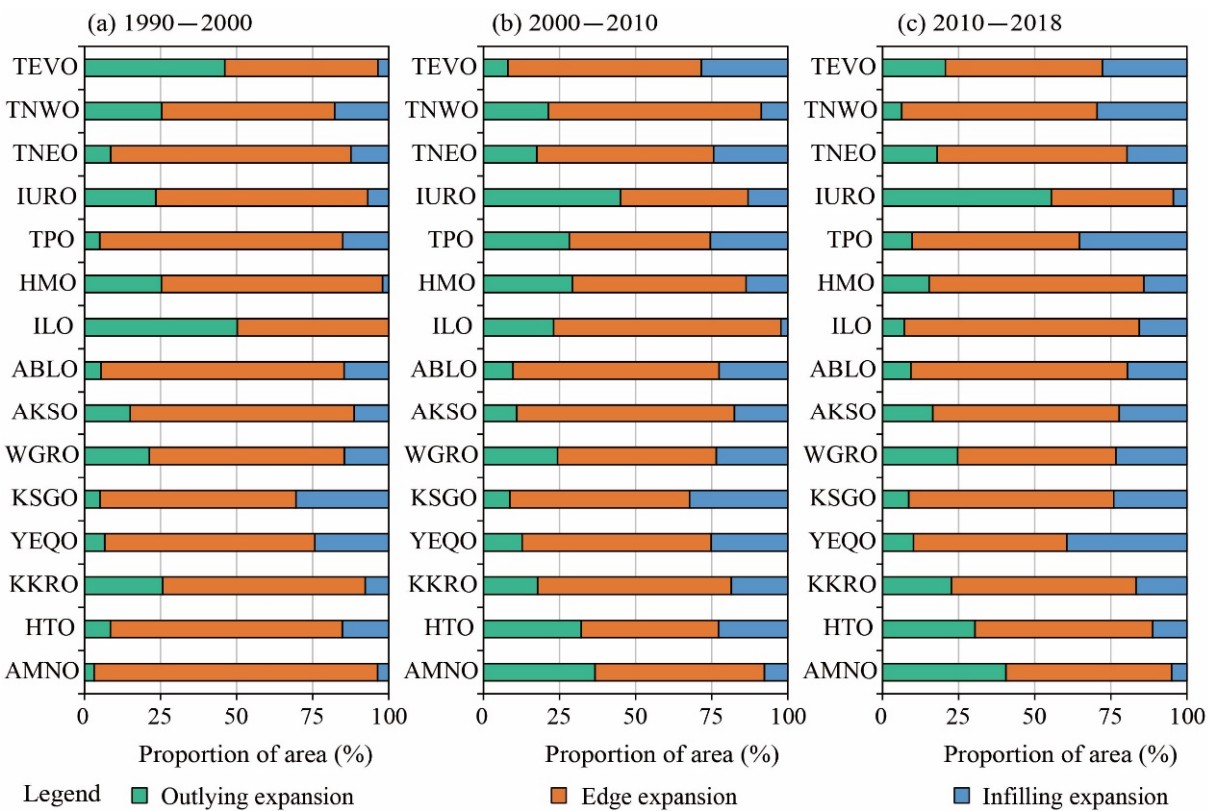

**Figure 8.** Evolution of the spatial patterns of cropland expansion in different oasis areas of Xinjiang during 1990–2018 (**a**–**c**).

**Table 5.** Statistics of land use conversion of cropland expansion in Xinjiang during 1990–2018.

| Period | Types of Land Use Conversion Modes for Cropland Expansion (%) | | | | |
| --- | --- | --- | --- | --- | --- |
| | Woodland (WL) | Grassland (GL) | Water Body (WB) | Built-Up Land (BL) | Unused Land (UL) |
| 1990–2000 | 1.01 | 92.04 | 0.11 | 0.09 | 6.75 |
| 2000–2010 | 0.48 | 88.91 | 0.40 | 0.00 | 10.20 |
| 2010–2018 | 1.65 | 76.58 | 0.38 | 0.37 | 21.02 |

Regarding the contribution rate of different land use conversion modes in different oasis areas to the new cropland (Figure 9), significant differences existed. Grassland occupation reached prominent contribution rates of cropland expansion in TNWO, AKSO, KKRO and YEQO, during the period 1990–2000. It recorded a relatively high contribution rate in IURO in addition to the four above-mentioned oasis areas during the period 2000–2010 and showed a strong polarization characteristic in TNWO. During the period 2010–2018, KSGO achieved the highest contribution rate of cropland expansion, followed

by TNWO, WGRO and AKSO. Regarding the mode of reclamation of unused land, it mainly occurred in AKSO during the period from 1990 to 2000. It made relatively big contributions to cropland expansion in TEVO, KKRO, TNWO and AKSO during 2000–2010. In addition, during 2010–2018, it registered the highest contribution rate of cropland expansion in IURO, and a relatively high contribution rate in HTO, KKRO and WGRO.

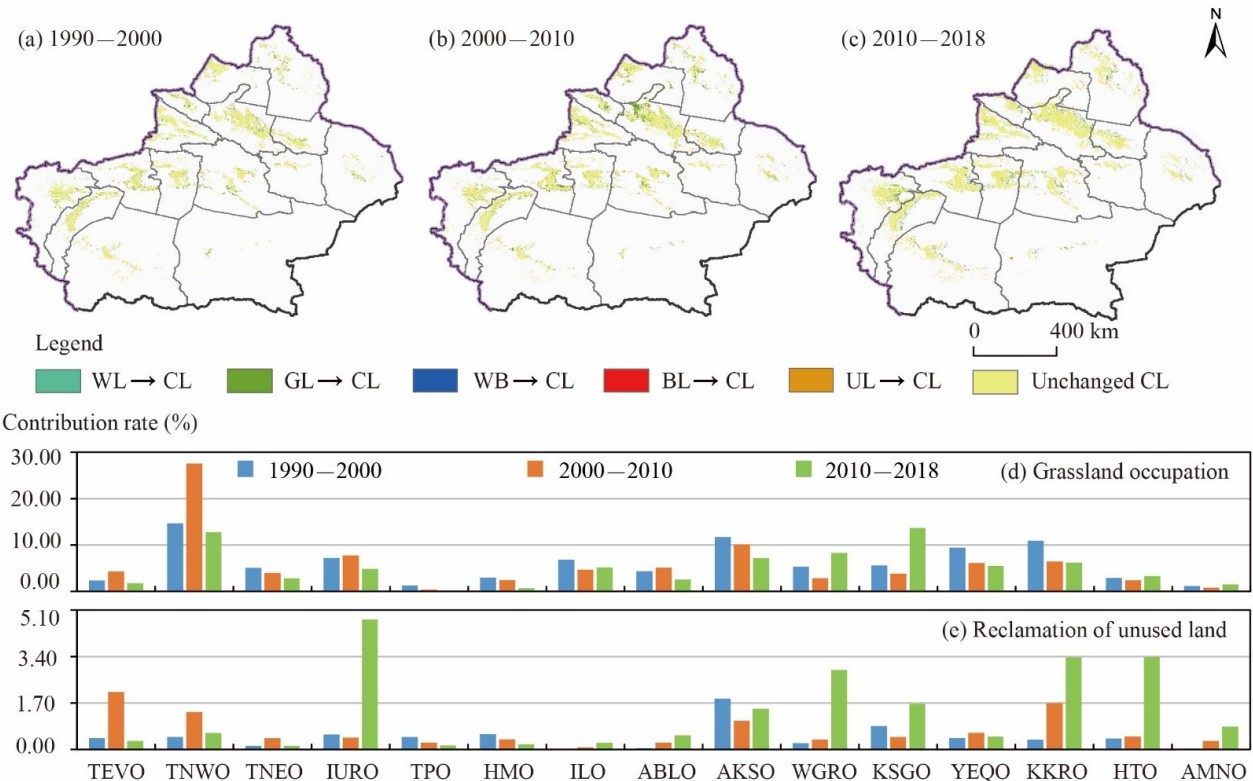

**Figure 9.** The spatial distribution of the land use conversion of cropland expansion in Xinjiang (**a**–**c**) and the changes in the contribution rate of each oasis area to the newly added cropland (**d**,**e**) during 1990–2018.

Regarding the evolution of the land use conversion modes of cropland expansion in different oasis areas (Figure 10), regional differences also existed to a certain extent. Similarly, a land use conversion mode was determined to be the dominant expansion mode in a period when the area attributable to this mode accounted for more than 33.33% of the total area. In general, land use conversion evolved in the following two scenarios. The first scenario was that the encroachment of grassland was the dominant expansion mode throughout the study period. This scenario was the case for an absolute majority of the oasis areas, including TEVO, TNWO, TNEO, HMO, ILO, ABLO, AKSO, WGRO, KSGO and YEQO. The second scenario was that the dominant land use conversion mode of cropland expansion experienced changes over different study periods. IURO, KKRO, HTO, AMNO and TPO were such examples.

### 3.3. Driving Forces of Cropland Expansion

3.3.1. The Scale of Xinjiang

Firstly, the nine influencing factors were discretized into four-level type variables in ArcGIS 10.7 software (West Redlands, CA, USA, see Table 6 for specific classification criteria). Secondly, the study area was divided into a gride with squares of 10 km × 10 km, and the area of new cropland in each grid square from 1990 to 2018 was calculated and used as a dependent variable; then, the sampling tool was used to extract the pixel values of variables and independent variables corresponding to the centroid points of the grids. Finally, the explanatory power (*q* value) of each influencing factor on the expansion of

cropland in Xinjiang was detected with the help of Geodetector software (http://www.geodetector.cn/, accessed on 20 May 2021).

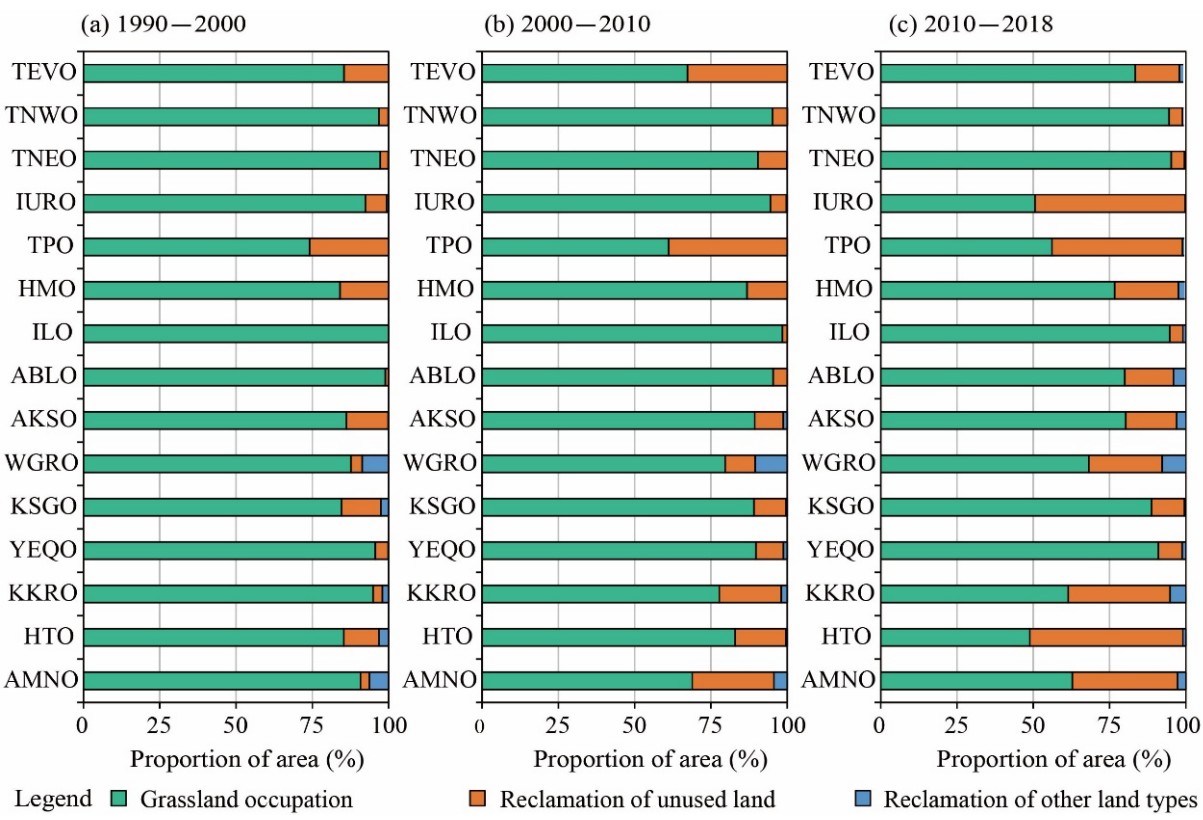

**Figure 10.** Evolution of the land use conversion modes of cropland expansion in different oasis areas in Xinjiang during 1990–2018 (**a**–**c**).

**Table 6.** Classification of factors affecting cropland expansion in Xinjiang.

| Factors | Unit | Classification | | | | Classification Basis |
| --- | --- | --- | --- | --- | --- | --- |
| | | I | II | III | IV | |
| ELV | m | <1000 | [1000, 2000) | [2000, 3000) | ≥3000 | [59] |
| SP | ° | <3 | [3, 8) | [8, 15) | ≥15 | [59] |
| AAP | mm | <100 | [100, 200) | [200, 400) | ≥400 | [59] |
| AT10 | °C | <800 | [800, 1600) | [1600, 3400) | ≥3400 | [59] |
| DRC | km | <5 | [5, 10) | [10, 20) | ≥20 | Expert advice [1] |
| TPOP | $10^4$ person | <2.01 | [2.01, 5.11) | [5.11, 10.75) | ≥10.75 | Natural Breaks |
| RPOP | $10^4$ person | <0.41 | [0.41, 1.61) | [1.61, 5.87) | ≥5.87 | Natural Breaks |
| PNIRR | CNY | <4158 | [4158, 5774) | [5774, 6763) | ≥6763 | Natural Breaks |
| PSTI | % | <13.59 | [13.59, 27.70) | [27.70, 38.51) | ≥38.51 | Natural Breaks |

[1] To obtain the division parameters in line with the actual situation of the study area, we distributed questionnaires to five experts in the field of oasis agriculture and sustainable development research in Xinjiang Institute of Ecology and Geography, Chinese Academy of Sciences. Based on the recommendations of these experts, we divided the DRC factor into four types of variables as shown in Table 5 using three parameters of 5 km, 10 km and 20 km.

According to the calculation results of the model (Table 7), it can be seen that the nine influencing factors all passed the 1% significance level test, indicating that the expansion of cropland in Xinjiang was under the influence of multiple factors. The explanatory power (*q* value) of each factor was ordered, and the ranking of the factors was: DRC (0.0642) > ELV (0.0308) > SP (0.0299) > TPOP (0.0275) > PNIRR (0.0260) > RPOP (0.0239) > AT10 (0.0218) > AAP (0.0180) > PSTI (0.0139). On the whole, cropland expansion in Xinjiang was primarily influenced by irrigation conditions, but at the same time also

subject to the comprehensive influences from topography, population growth and farmers' income. Hydrothermal conditions and industrial structure had relatively small influences on the expansion of cropland in the oasis areas.

**Table 7.** The power of different influencing factors to affect cropland expansion in Xinjiang based on Geodetector.

| Impact Factor | Geodetector Results | | | Impact Factor | Geodetector Results | | |
|---|---|---|---|---|---|---|---|
| | *p* Value | *q* Value | Explanatory Ranking | | *p* Value | *q* Value | Explanatory Ranking |
| ELV | 0.000 | 0.0308 | 2 | TPOP | 0.000 | 0.0275 | 4 |
| SP | 0.000 | 0.0299 | 3 | RPOP | 0.000 | 0.0239 | 6 |
| AAP | 0.000 | 0.0180 | 8 | PNIRR | 0.000 | 0.0260 | 5 |
| AT10 | 0.000 | 0.0218 | 7 | PSTI | 0.000 | 0.0139 | 9 |
| DRC | 0.000 | 0.0642 | 1 | — | — | — | — |

Irrigation conditions exerted a major influence on the expansion of cropland in the oasis areas due to low precipitation and large evaporation in arid areas. Water is the lifeline of irrigation agriculture. Although melt water from glaciers and mountain snow is the source of water for oasis agriculture, agricultural irrigation is mainly realized by diversion of water from natural rivers and artificial canals. The new cropland and irrigation facilities thus showed strong spatial coupling.

Topography indirectly affected the expansion of cropland through its influences on the difficulty of agricultural production and development. Different altitudes present different hydrothermal characteristics. On the one hand, the topographic relief determines the level of difficulty in cropland development, and on the other hand, it may also cause soil erosion during the use of cropland. In general, the topographic pattern of three mountains and two basins in Xinjiang determines that the expansion of cropland will mainly be located within the oasis areas and the transition zone between the oasis and the desert.

Population growth and farmers' incomes are important driving forces for the expansion of cropland in the oasis areas. Geographically far away from China's major grain production areas, Xinjiang has to solve its food security mainly through self-sufficiency. Growing populations lead to rising food demand, prompting people to cultivate more arable land resources to meet their basic needs. In addition, due to its relatively low level of urbanization and industrialization, rural residents in Xinjiang have to depend on cropland to engage in low-level agricultural production activities that serve as the main source of their employment and income.

### 3.3.2. Typical Oasis Areas

We selected the top six oasis areas (TNWO, AKSO, KKRO, IURO, KSGO and YEQO) in Xinjiang with the highest intensity in cropland expansion from 1990 to 2018 as typical areas to launch a geographical exploration of the explanatory power of the nine factors. According to the calculation results of the model (Figure 11), all the influencing factors of the six oasis areas passed the 10% significance level test, and most of them passed the 1% significance level test, indicating that cropland expansion in typical oasis areas during the study period was also under the influence of multiple factors, but the explanatory power of each influencing factor shows significant heterogeneity among different oasis areas.

On the whole, because the West, South and North sides of the KSGO are surrounded by high mountains, the topographical conditions (the *q* values of ELV and SP are 0.2141 and 0.2126, respectively) played a leading role in the expansion of cropland in this area. For the remaining five oasis areas, the factor of irrigation conditions (the *q* values of DRC of TNWO, AKSO, KKRO, IURO and YEQO are 0.1109, 0.1046, 0.1271, 0.0640 and 0.1137, respectively) still provided the dominant influence over their cropland expansion. In addition, topographical conditions also had a strong influence on the expansion of cropland in TNWO (the *q* values of ELV and SP are 0.0869 and 0.1064, respectively), YEQO (the *q* values of ELV and SP are 0.0726 and 0.1012, respectively) and IURO (the *q* values of

ELV and SP are 0.0306 and 0.0248, respectively). The influence of population growth on cropland expansion was more pronounced in KSGO (the *q* values of TPOP and RPOP are 0.1318 and 0.1422, respectively), YEQO (the *q* values of TPOP and RPOP are 0.0690 and 0.0522, respectively) and AKSO (the *q* values of TPOP and RPOP are 0.0685 and 0.0333, respectively). The population in these oasis areas has a large base, a high natural growth rate, and a low level of urbanization. During the study period, the total population and rural population in these oasis areas showed the characteristic of synchronous growth and large increment. The factor of farmers' income increase had a more prominent influence on the expansion of cropland in TNWO and KKRO (the *q* values of PNIRR are 0.0588 and 0.0795, respectively). These oasis areas have higher levels of agricultural productivity, and the purpose of cropland use has shifted from satisfying food demand to pursuit of high economic returns by growing more cash crops. The factor of industrial structure had a relatively significant influence on the expansion of cropland in YEQO and KSGO (the *q* values of PSTI are 0.0685 and 0.1128, respectively). This could be attributed mainly to the relatively high proportion of the primary industry in the national economic structure of these two oasis areas, so farmers have a strong dependence on cropland in both their production and life.

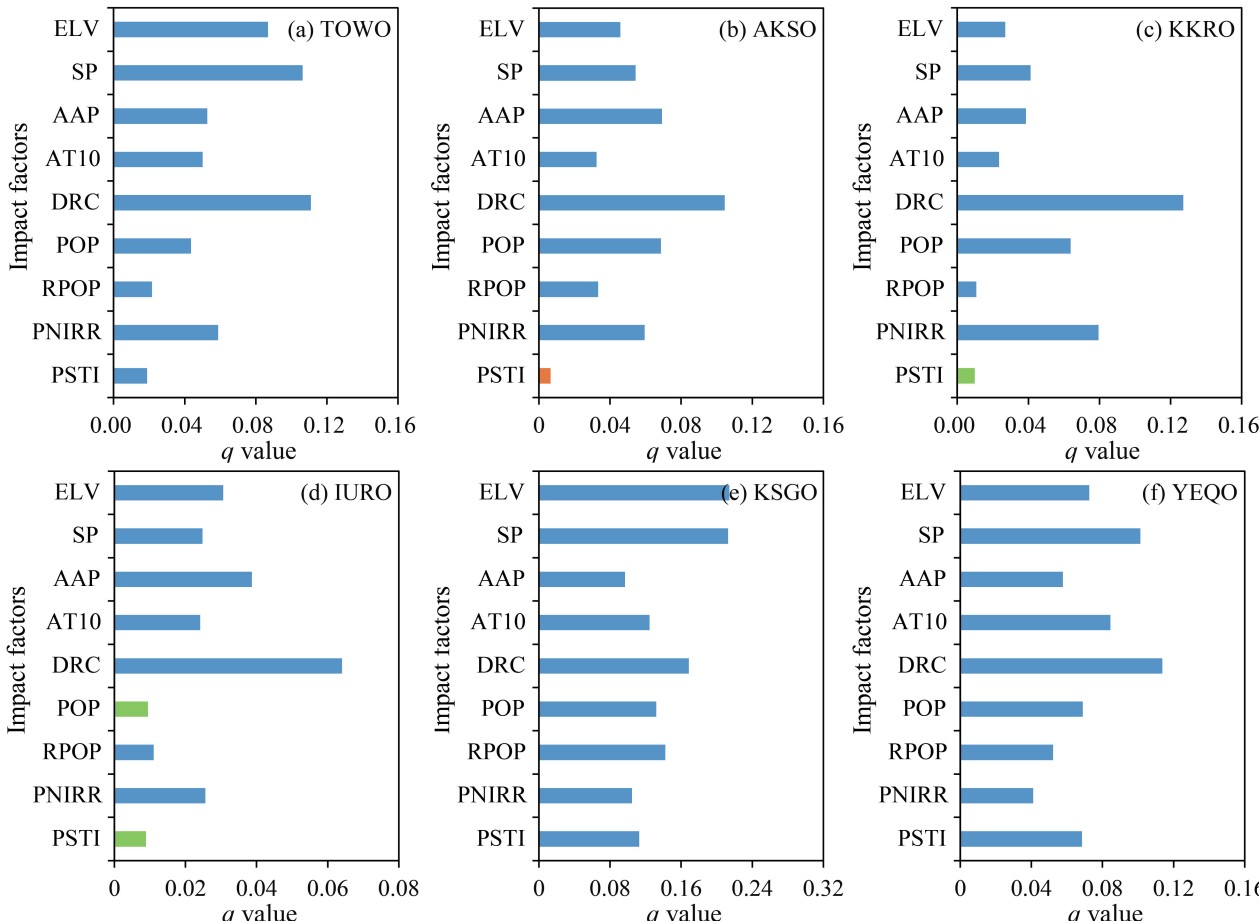

**Figure 11.** The power of different influencing factors to affect cropland expansion in typical oasis areas of Xinjiang (**a**–**f**) based on Geodetector. (Note: the colors blue, green, and orange indicate that the independent variable passed the significance test at the confidence level of 1%, 5%, and 10%, respectively.).

## 4. Discussion

### 4.1. Comparison of the Research Results of This Article with Results of Other Studies

The spatial data of cropland in this study come from the land use remote sensing monitoring data set released by the RESDC. The data set includes six primary types of land

use: cultivated land, forest land, grassland, water area, urban and rural construction land, and unused land, with a comprehensive evaluation accuracy of more than 93% [21,23]. That means that our conclusions reached on the basis of these data have good reliability.

Regarding the spatio-temporal process of cropland expansion, previous studies have found that cropland expansion in Xinjiang demonstrated obvious phases in the process, with the fastest expansion rate during 2000–2010 and the CGSDC moving southward on the whole [30,31,40], which were all confirmed in our research. When we updated the data and extended the study period to 2018, we found that the CLER and CLEI in Xinjiang began to slow down during the period 2010–2018, which was a new finding undiscovered by the abovementioned studies. This may be the result of the rigid restraint from the most stringent water resources management system on the expansion of cropland [60]. Using the "Qitai-Qira" line as the dividing line, we further found that the ratio of cropland in the southeast half of Xinjiang to the northwest half of Xinjiang remained basically stable at 1:9, showing strong dynamic stability. This is another new finding that is different from previous studies. The key factor causing this pattern of stable expansion is water. The northwest half has 93% of Xinjiang's water resources, while the southeast half has only 7% [58], which indirectly confirms the importance of water resources in the expansion of cropland in oasis areas. From the perspective of the interior of Xinjiang, we found that the expansion of cropland presented significant spatial heterogeneity. The TNWO had the highest intensity in cropland expansion, and the CLEI in IURO showed a continuous upward trend. In addition, AKSO, KKRO and KSGO in southern Xinjiang also had a high intensity of cropland expansion, while AMNO, HTO and HMO had a low expansion intensity, despite the rapid expansion of cropland in these areas. Compared with studies on the national scale [21–23] and on the overall scale of Xinjiang [29–31], our findings make up for the lack of research on spatial heterogeneity of cropland expansion in Xinjiang, which also demonstrates the necessity of this research.

Regarding the mode of cropland expansion, most scholars have performed analyses from the perspective of land use conversion and believe that the encroachment of grassland and the reclamation of unused land are the main methods of cropland expansion in Xinjiang [30,31,37]. Our research also reached similar conclusions. However, we were the first to examine the mode of cropland expansion in Xinjiang from the perspective of spatial pattern. We were amazed to find that edge expansion was the dominant mode of Xinjiang's cropland expansion, which showed a significant path dependence. The new cropland was mainly distributed along the periphery of the original cropland, and at the same time there was also a certain scale of infilling expansion and outlying expansion, indicating cropland expansion in Xinjiang showed characteristics of both agglomeration and dispersion.

Regarding the driving forces for the expansion of cropland, we integrated natural and socioeconomic factors into the Geodetector model and found that the dominant factor affecting the expansion of cropland in Xinjiang was irrigation conditions. It is undeniable that population growth and pursuit of interests are the main driving forces of global agricultural land expansion [6,10,13,15]. However, in arid regions, water, the lifeline of irrigated agriculture, decides the position of cropland distribution and the upper limit of its scale [25,49]. Without the support and guarantee of water resources, no matter how much land has been cultivated, it will eventually be degraded and abandoned. This is how oasis areas are different from other areas and this conclusion has also been supported by other studies on the driving forces of the cropland expansion in oasis areas [32,61,62]. Additionally, we found that the explanatory power of topography for cropland expansion in Xinjiang was also higher than that of population and economic factors. This is mainly due to the special topography of Xinjiang's mountains and basins, which has limited its cropland expansion. In previous research literature that did not take natural factors into consideration, factors such as population growth, technological progress, benefit induction, policy encouragement [41,45–47], etc., were identified as the main driving forces for the expansion of cropland in Xinjiang. However, when the impact of natural and socio-economic factors was taken into consideration, we found that socio-economic

factors exerted their influences on cropland expansion based on the natural environment background, which further demonstrates that cropland expansion is a product of the coupling of natural environment and human activities. This also suggests that we need to analyze the driving mechanism of cropland expansion in different regions according to the specific local conditions.

*4.2. The Ecological Risks of Cropland Expansion in Xinjiang and the Enlightenment about Sustainable Management*

During 1990–2018, the area of cropland in Xinjiang registered a net increase of 3135.65 thousand hectares, with the CLER of 1.93%. It is undeniable that the expanded scale of cropland has made a significant contribution to ensuring the security of Xinjiang's food supply, promoting the development of the agricultural economy, and supporting the employment of farmers and maintaining social stability. However, located in the center of the Eurasian continent, Xinjiang features an arid climate, a shortage of water resources, and a very fragile ecological environment. The high-intensity agricultural expansion has also brought severe ecological risks to the local area. On the one hand, agricultural development has exacerbated the stress on water resources in arid areas and changed the spatial distribution of water resources in river basins, leading to increasingly serious problems of desertification and salinization. Xinjiang's agricultural water consumption continued to increase from 44.1 billion $m^3$ in 1990 to 54.64 billion $m^3$ in 2015. As of 2015, agricultural water consumption accounted for 95% of Xinjiang's total water consumption (57.719 billion $m^3$), and Xinjiang's total water consumption has far exceeded Xinjiang's 2020 total water use control target (52.674 billion $m^3$) determined in "Implementation of the Assessment System for the Strict Water Resources Management System" issued by the State Council [63]. Spatially, the upper and middle oasis irrigation areas have caused continuous rise of groundwater levels due to large amounts of water diversion and flood irrigation. Compounded by the strong evaporation in arid areas, the problem of secondary salinization of farmland is severe. In the lower reaches of the river, due to the reduction of incoming water, the dry-up of the rivers, the decline of groundwater level, the desert vegetation has begun to degrade, leading to weakened ability in wind prevention and sand fixation, and the process of land desertification has been aggravated because of human behaviors [39,42,64]. On the other hand, the expansion of arable land in Xinjiang has worsened the fragile ecosystems in arid regions by occupying vast ecological land. Grassland and desert ecosystems in arid regions play an extremely important role in water conservation, wind prevention and sand fixation, soil and water conservation, carbon fixation and oxygen release, and biodiversity conservation [65–67]. However, grassland and unused land are the main sources of new cropland. This land conversion intensifies the trade-off effect between ecosystem service functions and poses a severe threat to the stability of the oases [35,36,68]. In fact, oasisization and desertification are two basic geographic processes that mutually affect each other in arid regions. The process of oasisization is manifested mainly in cropland expansion, but excessive cropland expansion will inevitably break the dynamic balance between oasis and desertification and then accelerate the process of desertification [25].

For these reasons, at the national level, the Chinese government must reconsider the dynamic balance of the total amount of cropland formed through the spatial transfer in the process of urbanization from the perspective of sustainable development, as this practice has contributed not only to the problem of spatial mismatch between cropland quality and food production capacity [69,70], but also to severe secondary ecological environmental problems in newly developed agricultural areas. It is not conducive to the realization of China's long-term national food security goals, and moreover, it contradicts the ecological civilization strategy that the Chinese government is vigorously promoting. Xinjiang plays an important strategic role in China's frontier stability, ethnic unity, and ecological security [58]. We should give full play to the multi-function synergy of cropland in maintaining social stability, promoting ethnic unity, promoting rural development, and

solidifying ecological protective screens in Xinjiang. This is the logic to be followed in the formulation of relevant policies at the national level.

At the local level, the key to sustainable management of oasis cropland in Xinjiang is to maintain a suitable cropland scale and promote the coordinated development of cropland, population, water resources, and industries. Specifically, firstly, we must actively respond to the ecological civilization strategy now being promoted by the Chinese government, with the permanent basic farmland protection red line system, the most stringent water resource management system, and the ecological protection red line system as rigid constraints [60], to strictly protect basic farmland, ensure the basic grain self-sufficiency, maintain an appropriate scale of cropland in accordance with the principle of "determining land by water", convert cropland that exceeds the carrying capacity of water resources into ecological land, and resolutely curb the encroachment of important ecological spaces. Secondly, we should transform the agricultural development method based on scale expansion and improve the quality and efficiency of cropland use by optimizing planting structure, increasing investment in science and technology, extending the agricultural industry chain, and cultivating green brands in accordance with local conditions. Thirdly, in the process of urbanization, we should guide the transfer of surplus rural labor to non-agricultural industries, strengthen vocational skills training, and broaden the channels for farmers to increase income, so as to weaken farmers' strong dependence on arable land at the source.

### 4.3. Merits, Limitations and Prospects

This research focused on Xinjiang, a typical region where the phenomenon of continuous expansion of cropland is occurring; the inverse of the process of rapid urbanization in China. Analysis in the research followed the research framework of "process-mode-driving force". We analyzed not only the phasal characteristics of the cropland expansion in Xinjiang, but also the spatial heterogeneity of cropland expansion in different oasis areas. In the description of spatial heterogeneity, we used the oasis area as the spatial unit, which well retained the characteristic of oasis cropland being continuously distributed in the watershed, while eliminating the problems presented by research on spatial difference performed solely on the basis of administrative divisions. In terms of expansion mode, we introduced the concept of spatial patterns, and comprehensively considered the two expansion modes of spatial patterns and land use conversion, so that we not only described the impact of cropland expansion on other land use types, but also revealed the characteristics of agglomeration and dispersion in the spatial locations of cropland expansion. In terms of driving forces, we not only considered the demographic, economic and other socioeconomic factors that have been popular focuses in the existing research, but also combined the actual characteristics of Xinjiang's geographic environment to take into account the factors of topography, geomorphology, hydrothermal conditions, and irrigation conditions, so as to achieve a more comprehensive understanding of the driving mechanism of cropland expansion. The conclusions of this study can provide a reference for Xinjiang to implement the strategy of ecological civilization and promote the rational development and use of cropland. At the same time, the research paradigm as provided by this article is also applicable to the study of the phenomenon of cropland expansion in other arid regions around the world.

However, this study also has limitations to a certain extent. On the one hand, Xinjiang has a special administrative management system. Two basic administration modes, "local jurisdiction" and "corps farm administration", have coexisted during the evolution of modern oases, so cropland expansion under the two different management modes will also be different in some ways [42]. However, due to the great difficulty of obtaining data on the administrative boundaries of the Corps, this study fails to distinguish between the two cases. Future research should pay more attention to the difference in the contributions to cropland expansion and the relevant formation mechanism under the two different modes of administrative management in Xinjiang, which will be an important basis for clarifying the responsibilities of cropland management under the two management modes.

On the other hand, existing studies have shown that the factor of policy has an important influence on the expansion of cropland [41,46,57,71]. However, this factor was not included in the analysis of driving forces in this research. This is because the Geodetector model we adopted requires the spatialization of the influencing factors and their discretization into type variables, and it is very difficult to achieve spatial quantification of the policy factor. In the future, we can try other methods to incorporate policy into the explanatory framework of the driving forces of cropland expansion to achieve a more scientific understanding of the driving mechanism of cropland expansion in oases.

## 5. Conclusions

This study takes Xinjiang in China as the case and uses the land use remote sensing monitoring data from 1990, 2000, 2010 and 2018 to comprehensively analyze the process characteristics, different modes, and driving mechanisms of cropland expansion in Xinjiang, as well as its spatial heterogeneity at the oasis area level. The research paradigm, as provided by this article, is also applicable to the study of the phenomenon of cropland expansion in other arid regions around the world. The main conclusions are as follows:

(1) Cropland in Xinjiang continued to expand, from 5802.89 thousand hectares in 1990 to 8938.54 thousand hectares in 2018, at the CLER of 1.93%, and experienced three stages of expansions: steady expansion, rapid expansion, and slow expansion. The shift of CGSDC in Xinjiang showed a characteristic of "slightly shifting toward southeast-leaping forward to northeast-moving back to southwest" in its trajectory. Divided by the "Qitai-Qira" line, the proportion of cropland in the southeast half and northwest half of Xinjiang stayed basically stable at 1:9, showing strong dynamic stability. The expansion of cropland in Xinjiang was dominated by edge expansion in terms of spatial patterns, while outlying expansion and infilling expansion coexisted. In terms of land use conversion, the expansion went through the transformation from a single mode dominated by encroachment on grassland to a dual development mode of encroachment on grassland and reclamation of unused land. The expansion of cropland in Xinjiang was affected by multiple factors. Irrigation conditions played a dominant role. Topography indirectly affected cropland expansion by affecting the suitability of agricultural production and development. Population growth and farmers' income were important driving forces.

(2) From the perspective of different oasis areas in Xinjiang, there was obvious spatial heterogeneity in the expansion of cropland. During 1990–2018, cropland expanded with a high intensity in TNWO, AKSO, KKRO, IURO and KSGO, and at a high rate in AMNO, IURO, ABLO, KKRO and AKSO. The trajectory of the shift of the CGSDC in the oasis areas approximately showed C-shaped, linear and circuitous trends of change. The edge expansion mode and grassland occupation mode of TNSO, AKSO, KKRO and KSGO contributed prominently to Xinjiang's new cropland. Topography was the dominant factor affecting the expansion of cropland in the KSGO, and irrigation conditions exerted the dominant influence in the other five typical oasis areas. In addition, the explanatory power of the population growth was more pronounced in the KSGO, YEQO and AKSO, while the explanatory power of the farmers' income increase was more obvious in the TNWO, AKSO and KKRO.

(3) The spatial shift of China's new cropland was not only detrimental to the realization of long-term national goals for food security, but also occupied a large amount of water resources and ecological land in Xinjiang and exacerbated the vulnerability of the ecosystem in arid regions. The key to sustainable management of cropland in Xinjiang in the future lies in maintaining an appropriate scale of cropland in accordance with the principle of "determining land by water" and promoting the coordinated development of cropland, population, water resources and industry.

**Author Contributions:** Conceptualization, T.C. and X.Z.; methodology, T.C., X.Z. and F.X.; software, J.Y.; validation, T.C.; formal analysis, F.X.; data curation, Z.Z.; writing—original draft preparation, T.C., X.Z., F.X., Z.Z., J.Y. and S.W.; visualization, S.W.; project administration, X.Z. All authors have read and agreed to the published version of the manuscript.

**Funding:** This research was funded by the CAS "Light of West China" Program (Grant No. 2019-XBQNXZ-A-005).

**Data Availability Statement:** Data on cropland come from the land use remote sensing monitoring data set re-leased by the Resource and Environment Science and Data Center of the Chinese Academy of Sciences (http://www.resdc.cn, accessed on 20 May 2021).

**Conflicts of Interest:** The authors declare no conflict of interest.

## Abbreviations

| Type | Item | Descriptions |
|---|---|---|
| Name of oasis district | TEVO | Tacheng-Emin Valley Oasis area |
| | TNWO | Tianshan North Slope West Oasis area |
| | TNEO | Tianshan North Slope East Oasis area |
| | IURO | Irtysh-Ulungu River Oasis area |
| | ILO | Ili Oasis area |
| | ABLO | Aibi Lake Oasis area |
| | TPO | Turpan Oasis area |
| | HMO | Hami Oasis area |
| | AKSO | Aksu Oasis area |
| | WGRO | Weigan River Oasis area |
| | KSGO | Kashgar Oasis area |
| | YEQO | Yerqiang Oasis area |
| | KKRO | Kaidu-Kongque River Oasis area |
| | HTO | Hotan Oasis area |
| | AMNO | Altun Mountain North Oasis area |
| Influencing factors | ELV | Elevation |
| | SP | Slope |
| | AAP | Annual average precipitation |
| | AT10 | >10 °C accumulated temperature |
| | DRC | Distance to rivers or canals |
| | TPOP | Total population |
| | RPOP | Rural population |
| | PNIRR | Per capita net income of rural residents |
| | PSTI | Proportion of secondary and tertiary industries |
| Others | TM | Thematic mapper |
| | OLI | Operational land imager |
| | CLNCA | cropland net change area |
| | CLER | cropland expansion rate |
| | CLEI | cropland expansion intensity |
| | CGSDC | the center of gravity of the spatial distribution of cropland |
| | LEI | the landscape expansion index |
| | RESDC | the Resource and Environment Science and Data Center of the Chinese Academy of Sciences |

## Appendix A

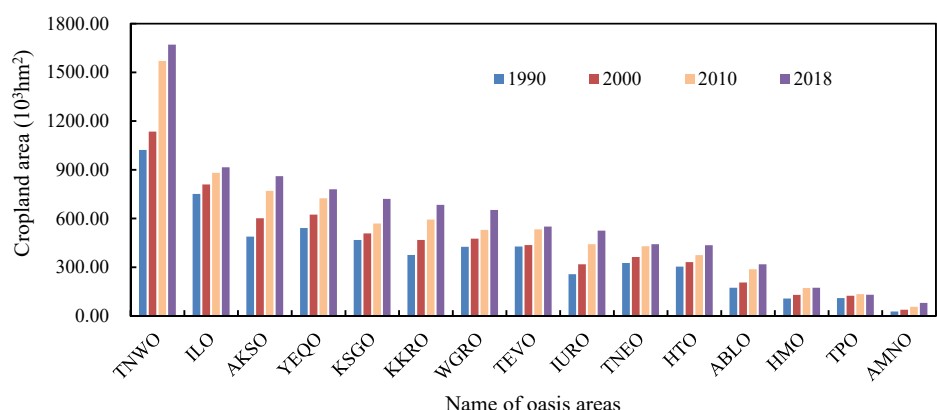

**Figure A1.** Changes in cropland area in each oasis area in Xinjiang during 1990–2018.

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
