# Peer review of "The Process-Mode-Driving Force of Cropland Expansion in Arid Regions of China Based on the Land Use Remote Sensing Monitoring Data"

_remotesensing, doi:10.3390/rs13152949_

Round 1
Reviewer 1 Report
This paper analyzes the cropland expansion during the period of 1990-2018 over the arid Xinjiang region in China using several interesting statistics mostly coming from urban applications. Below, please, find my major comments and suggestions:
1. The paper deals little with remote sensing data processing. In fact, another institution (Chinese Academy of Science) was responsible for producing the remote sensing based land use and land cover data used in this study. In this context, the MDPI´s Land journal is much more suitable than Remote Sensing journal for publishing this research.
2. I suggest adding “China” in the title. New title would be: The process-mode-driving force of cropland expansion in arid regions OF CHINA based on the land use remote sensing monitoring data.
3. Authors did a good job in the selection of the methods to describe the process and the mode of cropland expansion as well as to analyze the driving forces of cropland expansion. Congratulations! The Discussion section was also nicely conducted.
4. In Figure 3 and 5, please, identify the meanings of all short names in the title of the figure.
5. Figure A1 (land use types and spatial distribution in Xinjiang from 1990 to 2018) is a key figure of the manuscript. Therefore, I suggest moving to the main body of the text rather than presenting it as appendix.
If the editor decide to publish in the Remote Sensing journal, I rate this manuscript almost ready to be published.
Author Response
Thank you for your positive and constructive comments and suggestions on our manuscript. According to the recommendation, we have made careful modification in our manuscript. The reviewer’s suggestions are marked in blue, and our responses are marked in black. The detailed information can also be seen in our revised manuscript (with changes marked). We used the revision mode of the Word Software to modify the manuscript. The main corrections and the responds to the reviewer’s comments are as follows:
- The paper deals little with remote sensing data processing. In fact, another institution (Chinese Academy of Science) was responsible for producing the remote sensing based land use and land cover data used in this study. In this context, the MDPI´s Land journal is much more suitable than Remote Sensing journal for publishing this research.
R: Thanks for the suggestions. We believe that our manuscript is suitable for publication in Remote Sensing journal. We give the following two reasons.
On the one hand, Remote Sensing journal (ISSN 2072-4292) is relatively well-known journals in the field of remote sensing, although the goal of journals is to publish novel / improved methods / approaches and / or algorithms of remote sensing to benefit the community, open to everyone in need of them. But the research scope of the journal also involves all aspects of remote sensing science, including its application in geosciences, environmental sciences, ecology and civil engineering. Our research is about the expansion of cropland and its ecological risks in the oasis area of Xinjiang, China based on the land use remote sensing monitoring data in 1990, 2000, 2010 and 2018. This is consistent with Remote Sensing journal’s scope on remote sensing applications.
On the other hand, our research area (Xinjiang) is the largest provincial administrative region (with a total land area of approximately 1.66 million square kilometers) in China, accounting for one-sixth of China's land area. It is a huge and very difficult task to realize the interpretation of the fourth phase (1990, 2000, 2010 and 2018 year) of Xinjiang remote sensing image by oneself. Therefore, we chose to use existing remote sensing data products to carry out research on the expansion of cropland in Xinjiang. The spatial data of cropland in our manuscript comes directly from the land use remote sensing monitoring data set released by the Resource and Environment Science and Data Center of the Chinese Academy of Sciences (http://www.resdc.cn), with a spatial resolution of 30m and including land use data of 1990, 2000 and 2010 based on Landsat TM and the newly established land use data set of 2018 based on Landsat8 OLI. The comprehensive evaluation accuracy of the six classes of land use (cropland, woodland, grassland, water body, built-up land and unused land) was above 93%. Although we directly used the remote sensing data product and did not involve the specific processing of remote sensing data, this did not affect the quality of our research and the reliability of our conclusions. Because this set of data has been widely used in the field of land use cover and change at different geographic scales in China.
In summary, we believe that our manuscript is suitable for publication in remote sensing journals. In the later stage, we will consider selecting suitable research manuscripts for submission to MDPI's land journal.
- I suggest adding “China” in the title. New title would be: The process-mode-driving force of cropland expansion in arid regions OF CHINA based on the land use remote sensing monitoring data.
R: Thanks for the suggestion. We have added “of China” in the title. New title is: “The process-mode-driving force of cropland expansion in arid regions of China based on the land use remote sensing monitoring data”. The revised details can be found in Line 3, page 1 in the revised manuscript with changes marked.
- Authors did a good job in the selection of the methods to describe the process and the mode of cropland expansion as well as to analyze the driving forces of cropland expansion. Congratulations! The Discussion section was also nicely conducted.
R: Thank you very much for your affirmation of our manuscript, and we will continue to work hard.
- In Figure 3 and 5, please, identify the meanings of all short names in the title of the figure.
R: Thank you for your reminding. The abbreviations in Figure 3 and Figure 5 in our manuscript were formally defined when they first appeared in the text. Specifically, the definition of abbreviations in Figure 3 is detailed in line 307-333, page 7 in the revised manuscript with changes marked; the definition of abbreviations in Figure 5 (now is Figure 6) is detailed in line 182-189, page 7-8 in the revised manuscript with changes marked.
Due to the many abbreviations used in our manuscript, other reviewers also raised questions about the definition of abbreviations. In order to facilitate reviewers and readers to read our manuscript, we decided to add an explanatory sheet of all abbreviations in our manuscript. The explanatory sheet of all abbreviations (Table A1.) can be found in line 1145-1146, page 25 in the revised manuscript with changes marked.
Table A1. Abbreviations.
|
Type |
Item |
Descriptions |
|
Name of oasis district |
TEVO |
Tacheng-Emin Valley Oasis area |
|
|
TNWO |
Tianshan North Slope West Oasis area |
|
|
TNEO |
Tianshan North Slope East Oasis area |
|
|
IURO |
Irtysh -Ulungu River Oasis area |
|
|
ILO |
Ili Oasis area |
|
|
ABLO |
Aibi Lake Oasis area |
|
|
TPO |
Turpan Oasis area |
|
|
HMO |
Hami Oasis area |
|
|
AKSO |
Aksu Oasis area |
|
|
WGRO |
Weigan River Oasis area |
|
|
KSGO |
Kashgar Oasis area |
|
|
YEQO |
Yerqiang Oasis area |
|
|
KKRO |
Kaidu-Kongque River Oasis area |
|
|
HTO |
Hotan Oasis area |
|
|
AMNO |
Altun Mountain North Oasis area |
|
Influencing factors |
ELV |
Elevation |
|
|
SP |
Slope |
|
|
AAP |
Annual average precipitation |
|
|
AT10 |
>10℃ accumulated temperature |
|
|
DRC |
Distance to rivers or canals |
|
|
TPOP |
Total population |
|
|
RPOP |
Rural population |
|
|
PNIRR |
Per capita net income of rural residents |
|
|
PSTI |
Proportion of secondary and tertiary industries |
|
Others |
TM |
Thematic mapper |
|
|
OLI |
Operational land imager |
|
|
CLNCA |
cropland net change area |
|
|
CLER |
cropland expansion rate |
|
|
CLEI |
cropland expansion intensity |
|
|
CGSDC |
the center of gravity of the spatial distribution of cropland |
|
|
LEI |
the landscape expansion index |
|
|
RESDC
|
the Resource and Environment Science and Data Center of the Chinese Academy of Sciences |
- Figure A1 (land use types and spatial distribution in Xinjiang from 1990 to 2018) is a key figure of the manuscript. Therefore, I suggest moving to the main body of the text rather than presenting it as appendix.
R: We are very happy to adopt your advice. We have moved Figure A1(Land use types and spatial distribution in Xinjiang from 1990 to 2018) back to section 2.3 of the manuscript. The revised details can be found in Line 357-358, page 9 in the revised manuscript with changes marked.

Reviewer 2 Report
Consider following comments before publishing
- Line 86 - Provide abbreviation of TM and OLI
- Line 93 Gobi ? mean Gobi desert formed by Tibetan Plateau?
- Line 101 from Other Scholars - (but only one reference?)
- Source of the Figure 1?
- In figure 4 (c), there is some Chinese letter exist, please remove it.
- Line 510 mention the version of ArcGIS software.
- In table 5, DRC - Expert Advise, What does mean?Who gave advise, please mention.
Author Response
Thank you for your positive and constructive comments and suggestions on our manuscript. According to the recommendation, we have made careful modification in our manuscript. The reviewer’s suggestions are marked in blue, and our responses are marked in black. The detailed information can also be seen in our revised manuscript (with changes marked). We used the revision mode of the Word Software to modify the manuscript. The main corrections and the responds to the reviewer’s comments are as follows:
- Line 86 - Provide abbreviation of TM and OLI
R: Thank you for your reminding. This is due to our negligence. We have added the abbreviations for TM (thematic mapper) and OLI (operational land imager) in the revised manuscript with changes marked. It can be found in Line 89-90, page 2.
- Line 93 Gobi ? mean Gobi desert formed by Tibetan Plateau?
R: Thank you for your question. Your question may be caused by our translation process of the manuscript. Our original intention was to express the spatial expansion of cropland from the alluvial plain with good water and soil resource conditions in the oasis to the desert beyond the oasis. In fact, the word "desert" can fully express what we mean. Therefore, we decided to delete the word "Gobi" from our manuscript. Thank you again for your reminding. The revised details can be found in Line 94-95, page 2 in the revised manuscript with changes marked.
- Line 101 from Other Scholars - (but only one reference?)
R: Thank you for your reminding. This is a mistake caused by our carelessness. It is true that only one scholar’s literature is cited here, and there is indeed a contradiction in the use of "other scholars" to describe this. We have corrected this error in the manuscript. The revised details can be found in Line 110, page 3 in the revised manuscript with changes marked.
- Source of the Figure 1?
R: Thank you for your question. Figure 1 is an overview map of the geographic location of Xinjiang and its oasis division. All data sources used in Figure 1 are detailed in section 2.3 of the manuscript. Specifically, the vector data of administrative boundaries and rivers come from the 1:1 million Chinese basic geographic database provided by the China’ National Catalogue Service for Geographic Information (http://www.webmap.cn/). The DEM data come from Geospatial Data Cloud (http://www.gscloud.cn), with a spatial resolution of 30m. The division scheme of Xinjiang Oasis refers to Han Delin's monograph (Artificial Oasis in Xinjiang), which has been explained in the section 2.1 of our manuscript.
- In figure 4 (c), there is some Chinese letter exist, please remove it.
R: Thank you for your reminding. We are very sorry for our carelessness. We have modified Figure 4(c) (now is Figure 5(c)). The revised details can be found in Line 434-435, page 12 in the revised manuscript with changes marked.
- Line 510 mention the version of ArcGIS software.
R: Thank you for your reminding. We have supplemented the version of the ArcGIS software in the revised manuscript.
- In table 5, DRC - Expert Advise, What does mean? Who gave advise, please mention.
R: Thanks for the suggestion. In our manuscript, we used the Geodetector model to analyze the differences in driving forces for the expansion of cropland in Xinjiang. According to the needs of the model, the influencing factors need to be treated as type variables before calculation. Irrigation conditions are the basic guarantee for the development of oasis agriculture and an important factor influencing the expansion of cropland. Agricultural irrigation is realized by the diversion of water from natural rivers or artificial canals. We used the spatial distance to rivers or canals (DRC) to describe the irrigation conditions.
Dividing DRC factor into type variables, this step needs to be completed in conjunction with the actual situation of the study area. In order to obtain the division parameters in line with the actual situation of the study area, we distributed questionnaires to five experts in the field of oasis agriculture and sustainable development research in Xinjiang Institute of Ecology and Geography, Chinese Academy of Sciences, and consulted them for their suggestions on how to treat DRC factor as typological variables. Finally, based on the recommendations of these experts, we divided the DRC factor into four types of variables as shown in Table 5 using three parameters of 5 km, 10 km and 20 km. In order to eliminate the reader's confusion and facilitate understanding, we have added a note to Table 5 (now is Table 6) to explain this problem. The revised details can be found in Line 816-820, page 18 in the revised manuscript with changes marked.

Reviewer 3 Report
This manuscript presents an interesting approach for examining what factors influence cropland expansion in arid regions. However, it has some unclear issues. Please see the following list of comments and suggestions:
Title: The title might benefit from including the geographical location.
- What are the innovative contributions of your manuscript to science? This is not clear as there are a lot of articles that analyse these issues.
- The methodology needs more explanations regarding alternative approaches. Why did you use techniques/methods such as Geodetector, landscape expansion index, or Markov land-use transition matrix? instead of using other techniques/methods to understand and analyse the dynamics of land-use change? This needs to be well justified.
Line 156: “….with an urbanization rate of 50.91%” from when to when?
Line 277-279: Why these factors and not others? Have you performed any multicollinearity tests?
Section 3 may appear in a table, with columns indicating the name of the data, year, original resolution, explanation of the variable, data source, etc.
- I don't understand why you wrote:
“4.1. Comparison of the research results of this article with results of other studies”
“4.2. Comparison of the research results of this article with results of other studies”
- Minor grammar and punctuation errors can be found throughout the text and need to be corrected.
Author Response
Thank you for your positive and constructive comments and suggestions on our manuscript. According to the recommendation, we have made careful modification in our manuscript. The reviewer’s suggestions are marked in blue, and our responses are marked in black. The detailed information can also be seen in our revised manuscript (with changes marked). We used the revision mode of the Word Software to modify the manuscript. The main corrections and the responds to the reviewer’s comments are as follows:
- Title: The title might benefit from including the geographical location.
R: Thank you for your advices, we have refinished the title of the manuscript. The new title is: “The process-mode-driving force of cropland expansion in arid regions of China based on the land use remote sensing monitoring data”. The revised details can be found in Line 3, page 1 in the revised manuscript with changes marked.
- What are the innovative contributions of your manuscript to science? This is not clear as there are a lot of articles that analyse these issues.
R: Thank you for your suggestion. The center of gravity of China's new cropland has shifted from Northeast China to the Xinjiang oasis areas where the ecological environment is relatively fragile. However, we currently face a lack of a comprehensive review of the cropland expansion in oasis areas of Xinjiang, which is importantly associated with the sustainable use of cropland, social stability and oasis ecological security. Although some scholars have studied the expansion of cropland in Xinjiang from different geographic scales, there are still some shortcomings. Based on the existing research foundation and deficiencies, we conducted an in-depth and systematic study on the phenomenon of cropland expansion from the whole Xinjiang and its different oasis areas.
The differences between our research and previous research can be summarized in the following four points. First, we constructed a research framework of cropland expansion based on the process-mode-driving force paradigm. Second, we conducted an in-depth discussion on the spatial heterogeneity of cropland expansion from the scale of 15 oasis areas in Xinjiang. Third, we used the landscape expansion index for the first time to quantify and analyze the spatial pattern of cropland expansion. Fourth, we comprehensively analyzed how the factors of natural environment and socio-economy have served as the driving forces of cropland expansion in Xinjiang as well as its spatial heterogeneity.
Regarding the specific research conclusions, some of our conclusions are consistent with existing research findings, and there are also some new findings that are different from existing research. These comparative analyses are detailed in section 4.1 of our manuscript. In addition, we discussed the ecological risks that may be brought by the cropland expansion in Xinjiang and proposed suggestions for sustainable management of cropland in the future. We believe that our research is of great significance for comprehensively and objectively understanding the phenomenon of cropland expansion in Xinjiang. The conclusions of this paper can provide a reference for Xinjiang to implement the strategy of ecological civilization and promote the rational development and utilization of cropland. At the same time, the research paradigm as provided by this article is also applicable to the study of the phenomenon of cropland expansion in other arid regions around the world.
- The methodology needs more explanations regarding alternative approaches. Why did you use techniques/methods such as Geodetector, landscape expansion index, or Markov land-use transition matrix? instead of using other techniques/methods to understand and analyse the dynamics of land-use change? This needs to be well justified.
R: Thank you for your suggestion. The research methods we used come from the existing literature. When making specific choices, we also combine the advantages of the methods, the maturity of the current use, and the needs of our research. Based on your suggestion, we have strengthened the description of the method part in the revised manuscript. The revised details can be found in Line 199 of page 5, in Line 245-247 of page 6, in Line 262-266 of page 6 and in Line 278-281 of page 7 in the revised manuscript with changes marked.
- Line 156: “….with an urbanization rate of 50.91%” from when to when?
R: Thank you for your reminding. The time point for the urbanization rate of 50.91% is 2018. The original description of the data in our manuscript is as follows: “As of 2018, Xinjiang had 8.94 million hectares of arable land which accounted for about 5% of the total land area of Xinjiang, and a total population of 24.87 million people with an urbanization rate of 50.91%”. This can be found in Line 166-168, page 4 in the revised manuscript with changes marked.
- Line 277-279: Why these factors and not others? Have you performed any multicollinearity tests?
R: Thank you for your question. Regarding why these factors that affect the expansion of cropland were selected, we have made a detailed explanation in the second part (selection of influencing factors and definition of variables) in section 2.2.3 of the original manuscript. There are three main principles for our selection: one is to learn from the existing literature research results, the other is to combine Xinjiang's actual natural and socio-economic conditions, and the third is to consider the availability of data. On this basis, we finally determined nine factors affecting the expansion of cropland in Xinjiang from six levels (including topography, hydrothermal conditions, irrigation conditions, population growth, farmers’ income increase, and industrial structure). Of course, due to the inherent requirements of the Geodetector method and the availability of data, there are also some influencing factors that have not been considered. We also discussed these shortcomings and future improvements in section 4.3 of our manuscript.
Regarding the question of whether the multicollinearity tests of influencing factors has been done, our answer is not. Because we used the Geodetector model (http://www.geodetector.org/) in the manuscript to analyze the explanatory power of influencing factors. Geodetector is a statistical method to detect spatial stratified heterogeneity and reveal the driving factors behind it. Its core idea is: if an independent variable has an important influence on a dependent variable, then the spatial distribution of the independent variable and the dependent variable should be similar. The geographic detector essentially reveals the relationship between X and Y in the spatial distribution, and does not involve the relationship between X and X, so there is no need to perform multicollinearity tests between independent variables.
- Section 3 may appear in a table, with columns indicating the name of the data, year, original resolution, explanation of the variable, data source, etc.
R: Thank you for your suggestions. We have added a table on data sources in section 2.3 of our manuscript based on your comments. This (Table 1.) can be found in Line 360, page 10 in the revised manuscript with changes marked.
- I don't understand why you wrote:“4.1. Comparison of the research results of this article with results of other studies”“4.2. Comparison of the research results of this article with results of other studies”
R: Thank you for your reminding. This is an error caused by our carelessness. The correct title of section 4.2 in our manuscript should be " The ecological risks of cropland expansion in Xinjiang and the enlightenment about sustainable management ". We have corrected this error. This can be found in Line 970-971, page 21 in the revised manuscript with no changes marked.
- Minor grammar and punctuation errors can be found throughout the text and need to be corrected.
R: Thank you for your reminding. We have done a language retouching of all the content of our manuscript through the professional language service company. The revised details can be found in the revised manuscript with changes marked.

Reviewer 4 Report
This article seems to be correctly arranged and clearly shows the applied method of collecting data on land use changes. The cartographic part of figures 4, 6, 8, and A1 does not show differences in the spatial distribution between the epochs, due to the small scale. The paper does not sufficiently analyze the remote sensing methods, which were used. In the chapter conclusion, can also be mentioned something about the quality of the applied method.
Author Response
Thank you for your positive and constructive comments and suggestions on our manuscript. According to the recommendation, we have made careful modification in our manuscript. The reviewer’s suggestions are marked in blue, and our responses are marked in black. The detailed information can also be seen in our revised manuscript (with changes marked). We used the revision mode of the Word Software to modify the manuscript. The main corrections and the responds to the reviewer’s comments are as follows:
- This article seems to be correctly arranged and clearly shows the applied method of collecting data on land use changes. The cartographic part of figures 4, 6, 8, and A1 does not show differences in the spatial distribution between the epochs, due to the small scale.
R: Thank you for your suggestion. We have adjusted the scale of the above four pictures according to your suggestions. Since another reviewer suggested that Figure A1 should be moved to the main text, the serial number of the new figure in the revised manuscript has also changed. The revised details can be found in Line 357-358 of page 9 (Figure 4), in Line 434-435 of page 12 (Figure 5), in Line 572 of page 14 (Figure 7) and in Line 738 of page 17 (Figure 9) in the revised manuscript with changes marked.
- The paper does not sufficiently analyze the remote sensing methods, which were used.
R: Thank you for your suggestion. Our manuscript is a study on the process-mode-driving force of cropland expansion in the oasis area of Xinjiang, China based on the land use remote sensing monitoring data in 1990, 2000, 2010 and 2018. Xinjiang is the largest province in China (with a total land area of approximately 1.66 million square kilometers), accounting for one-sixth of China's land area. It is a huge and very difficult task to realize the interpretation of the fourth phase (1990, 2000, 2010 and 2018 year) of Xinjiang remote sensing image by oneself. Therefore, we chose to use existing remote sensing data products to carry out research on the expansion of cropland in Xinjiang.
The spatial data of cropland in our manuscript comes directly from the land use remote sensing monitoring data set released by the Resource and Environment Science and Data Center of the Chinese Academy of Sciences (http://www.resdc.cn), with a spatial resolution of 30m and including land use data of 1990, 2000 and 2010 based on Landsat TM and the newly established land use data set of 2018 based on Landsat8 OLI. The comprehensive evaluation accuracy of the six classes of land use (cropland, woodland, grassland, water body, built-up land and unused land) was above 93%. Although we directly used the remote sensing data product and did not involve the specific processing of remote sensing data, this did not affect the quality of our research and the reliability of our conclusions. Because, on the one hand, this set of data is released by an authoritative scientific research institution in China; on the other hand, this set of data has been widely used in the field of land use cover and change at different geographic scales in China.
- In the chapter conclusion, can also be mentioned something about the quality of the applied method.
R: Thank you for your suggestion. Since we directly used the existing remote sensing data products to carry out the research, we put the quality of the data and the reliability of the research conclusions in Section 4.1 for discussion, and we did not mention these in the conclusion. According to your suggestion, we have added a new content about the reproducibility of our research method in the chapter conclusion, and its specific expression is:” The research paradigm as provided by this article is also applicable to the study of the phenomenon of cropland expansion in other arid regions around the world.” The revised details can be found in Line 1092-1094, page 23 in the revised manuscript with changes marked.

Reviewer 5 Report
The Article “The Process-Mode-Driving Force of Cropland Expansion in Arid Regions Based on the Land Use Remote Sensing Monitoring Data” submitted by Tianyi Cai 1,2, Xinhuan Zhang et al is a valuable work and can be published after some minor corrections:
In the paper, the authors use a lot of abbreviations and for a reader is difficult to follow the results. We consider that a more accessible form should be found, for example, you can use: Tacheng instead of (TEVO), Tianshan NW instead of (TNWO), Tianshan NEn(TNEO), Irtysh -Ulungu (IURO), Ili instead of (ILO) and Aibi Lake instead of (ABLO); For the Southern oasis area Turpan instead of (TPO), Hami instead of (HMO), Aksu instead of (AKSO), Weigan instead of (WGRO), Kashgar instead of (KSGO), Yerqiang instead of (YEQO), Kaidu-Kongque instead of (KKRO), Hotan instead of (HTO) and Altun instead of (AMNO). These suggestions are not mandatory, the authors decide what they use, but using the same form of the abbreviation for the regions and for the indices with capital letters transform the text into a difficult text and all the discoveries are hidden in this difficulty. Also, the indices the author use could be explained in the text, that preceded the images or graphs. For example from line 560 until 590, there is a part of the text full of abbreviations and if you are not used to these terms, you will have a lot of difficulties in understanding the main idea.
Also, I consider that the land cover maps resulted from the interpretation of the satellite images should be presented in a larger version, they are too small. It is impossible to spot the differences between the years if the images are reduced too much. The study region is large and you can not choose to show 3 different situations in a word row…..it looks like Thumbnails.
I am not a native of English but please check some possible problems:
Line 769 - ) Cropland in Xinjiang continued to expanded ( probably you wanted to write expand)from 5,802.89 thousand hectares in 769 1990 to 8,938.54 thousand hectares in 2018 at the CLER of 1.93%
Also, we believe that in current language the syntagm “With respect to the…” is less used than ”Regarding the…” please verify, the language of the article is clear and well written, so these could be minor errors, or my mistakes.
Thank You!
Author Response
Thank you for your positive and constructive comments and suggestions on our manuscript. According to the recommendation, we have made careful modification in our manuscript. The reviewer’s suggestions are marked in blue, and our responses are marked in black. The detailed information can also be seen in our revised manuscript (with changes marked). We used the revision mode of the Word Software to modify the manuscript. The main corrections and the responds to the reviewer’s comments are as follows:
- In the paper, the authors use a lot of abbreviations and for a reader is difficult to follow the results. We consider that a more accessible form should be found, for example, you can use: Tacheng instead of (TEVO), Tianshan NW instead of (TNWO), Tianshan NEn(TNEO), Irtysh -Ulungu (IURO), Ili instead of (ILO) and Aibi Lake instead of (ABLO); For the Southern oasis area Turpan instead of (TPO), Hami instead of (HMO), Aksu instead of (AKSO), Weigan instead of (WGRO), Kashgar instead of (KSGO), Yerqiang instead of (YEQO), Kaidu-Kongque instead of (KKRO), Hotan instead of (HTO) and Altun instead of (AMNO). These suggestions are not mandatory, the authors decide what they use, but using the same form of the abbreviation for the regions and for the indices with capital letters transform the text into a difficult text and all the discoveries are hidden in this difficulty. Also, the indices the author use could be explained in the text, that preceded the images or graphs. For example from line 560 until 590, there is a part of the text full of abbreviations and if you are not used to these terms, you will have a lot of difficulties in understanding the main idea.
R: Thank you for your suggestion. We admit that we have used many abbreviations in our manuscript, including oasis area, influencing factors, and other abbreviations. Since these proprietary names are repeated many times in the manuscript, it is obviously not in line with the requirement that the language should be concise if the full expression is used for each occurrence. In order to solve this problem, we made a comprehensive definition of these proprietary names when they first appeared in the manuscript, and then used abbreviations when they appeared again. This has simplified the length and volume of the manuscript, but also brought some reading difficulties to readers.
As for the modification plan for the abbreviation of the Oasis area you recommended, we first express our gratitude. We thought about your plan before, but there are indeed some problems. The geographic scope of the Xinjiang Oasis Division adopted in our manuscript is not consistent with the geographic scope of Xinjiang’s prefectural administrative divisions. For example, Tacheng-Emin Valley Oasis area (TEVO) is composed of Tacheng City, Emin County, Yumin County, Toli County and Hoboksar Mongol Autonomous County. However, in addition to the above five counties, the Tacheng Prefecture also includes Usu City and Shawan County. Therefore, if we directly use Tacheng instead of the Tacheng-Emin Valley Oasis area (TEVO), it is easy for readers to misunderstand the geographical scope. Similar misunderstandings also occur in oasis areas such as Yili, Aksu, and Hotan.
In the end, we adopted the opinions of the other two reviewers and added a full-text abbreviation table to the manuscript attachment to help readers better understand the content of the article. The explanatory sheet of all abbreviations (Table A1.) can be found in line 1148, page 25 in the revised manuscript with changes marked.
Table A1. Abbreviations.
|
Type |
Item |
Descriptions |
|
Name of oasis district |
TEVO |
Tacheng-Emin Valley Oasis area |
|
|
TNWO |
Tianshan North Slope West Oasis area |
|
|
TNEO |
Tianshan North Slope East Oasis area |
|
|
IURO |
Irtysh -Ulungu River Oasis area |
|
|
ILO |
Ili Oasis area |
|
|
ABLO |
Aibi Lake Oasis area |
|
|
TPO |
Turpan Oasis area |
|
|
HMO |
Hami Oasis area |
|
|
AKSO |
Aksu Oasis area |
|
|
WGRO |
Weigan River Oasis area |
|
|
KSGO |
Kashgar Oasis area |
|
|
YEQO |
Yerqiang Oasis area |
|
|
KKRO |
Kaidu-Kongque River Oasis area |
|
|
HTO |
Hotan Oasis area |
|
|
AMNO |
Altun Mountain North Oasis area |
|
Influencing factors |
ELV |
Elevation |
|
|
SP |
Slope |
|
|
AAP |
Annual average precipitation |
|
|
AT10 |
>10℃ accumulated temperature |
|
|
DRC |
Distance to rivers or canals |
|
|
TPOP |
Total population |
|
|
RPOP |
Rural population |
|
|
PNIRR |
Per capita net income of rural residents |
|
|
PSTI |
Proportion of secondary and tertiary industries |
|
Others |
TM |
Thematic mapper |
|
|
OLI |
Operational land imager |
|
|
CLNCA |
cropland net change area |
|
|
CLER |
cropland expansion rate |
|
|
CLEI |
cropland expansion intensity |
|
|
CGSDC |
the center of gravity of the spatial distribution of cropland |
|
|
LEI |
the landscape expansion index |
|
|
RESDC
|
the Resource and Environment Science and Data Center of the Chinese Academy of Sciences |
- Also, I consider that the land cover maps resulted from the interpretation of the satellite images should be presented in a larger version, they are too small. It is impossible to spot the differences between the years if the images are reduced too much.
R: Thank you for your suggestion. We have adjusted the size of Figure 4, Figure 5, Figure 7 and Figure 9 based on your comments and presented them in a larger version. The revised details can be found in Line 357-358 of page 9 (Figure 4), in Line 434-435 of page 12 (Figure 5), in Line 572 of page 14 (Figure 7) and in Line 738 of page 17 (Figure 9) in the revised manuscript with no changes marked.
- The study region is large and you can not choose to show 3 different situations in a word row…..it looks like Thumbnails.
R: Thank you for your reminding. We have modified the display content of the research area overview map. The revised details can be found in Line 192, page 4 in the revised manuscript with changes marked.
- I am not a native of English but please check some possible problems: Line 769 - ) Cropland in Xinjiang continued to expanded ( probably you wanted to write expand)from 5,802.89 thousand hectares in 769 1990 to 8,938.54 thousand hectares in 2018 at the CLER of 1.93%。Also, we believe that in current language the syntagm “With respect to the…” is less used than ”Regarding the…” please verify, the language of the article is clear and well written, so these could be minor errors, or my mistakes.
R: Thank you for your reminding. We have modified this error in the revised manuscript. In addition, we have done a language retouching of all the content of our manuscript through the professional language service company. The revised details can be found in the revised manuscript with changes marked.

Round 2
Reviewer 3 Report
Thank you and your colleagues for the changes that you have made to this manuscript and how well you have answered the comments and suggestions.